# Integrated miRNA Profiling of Extracellular Vesicles from Uterine Aspirates, Malignant Ascites and Primary-Cultured Ascites Cells for Ovarian Cancer Screening

**DOI:** 10.3390/pharmaceutics16070902

**Published:** 2024-07-05

**Authors:** Gleb O. Skryabin, Andrei V. Komelkov, Kirill I. Zhordania, Dmitry V. Bagrov, Adel D. Enikeev, Sergey A. Galetsky, Anastasiia A. Beliaeva, Pavel B. Kopnin, Andey V. Moiseenko, Alexey M. Senkovenko, Elena M. Tchevkina

**Affiliations:** 1N.N. Blokhin National Medical Research Center of Oncology, 24 Kashirskoye Highway, Moscow 115522, Russia; g.skrjabin@ronc.ru (G.O.S.); camel1000@yandex.ru (A.V.K.); kiazo2@yandex.ru (K.I.Z.); adelbufyeni@mail.ru (A.D.E.); s.galetskiy@ronc.ru (S.A.G.); belka595@gmail.com (A.A.B.); pbkopnin@mail.ru (P.B.K.); 2Faculty of Biology, Lomonosov Moscow State University, 1-12 Leninskie Gory, Moscow 119991, Russia; dbagrov@gmail.com (D.V.B.); postmoiseenko@gmail.com (A.V.M.); senkoven99@gmail.com (A.M.S.)

**Keywords:** miRNA, ovarian cancer, uterine aspirates, extracellular vesicles, liquid biopsy, differential expression, malignant ascites

## Abstract

Extracellular vesicles (EVs) are of growing interest in the context of screening for highly informative cancer markers. We have previously shown that uterine aspirate EVs (UA EVs) are a promising source of ovarian cancer (OC) diagnostic markers. In this study, we first conducted an integrative analysis of EV-miRNA profiles from UA, malignant ascitic fluid (AF), and a conditioned medium of cultured ascites cells (ACs). Using three software packages, we identified 79 differentially expressed miRNAs (DE-miRNAs) in UA EVs from OC patients and healthy individuals. To narrow down this panel and select miRNAs most involved in OC pathogenesis, we aligned these molecules with the DE-miRNA sets obtained by comparing the EV-miRNA profiles from OC-related biofluids with the same control. We found that 76% of the DE-miRNAs from the identified panel are similarly altered (differentially co-expressed) in AF EVs, as are 58% in AC EVs. Interestingly, the set of miRNAs differentially co-expressed in AF and AC EVs strongly overlaps (40 out of 44 miRNAs). Finally, the application of more rigorous criteria for DE assessment, combined with the selection of miRNAs that are differentially co-expressed in all biofluids, resulted in the identification of a panel of 29 miRNAs for ovarian cancer screening.

## 1. Introduction

Worldwide, ovarian cancer (OC) remains one of the leading causes of gynecologic cancer mortality. Most OC patients are diagnosed with advanced-stage cancer, resulting in a five-year survival rate of less than 30%, and there are currently no effective early detection strategies [1]. Therefore, the search for markers for early non-invasive diagnosis of OC is one of the most important tasks in gynecologic oncology.

Extracellular vesicles (EVs) are a heterogeneous group of secreted membrane-bound particles that play a role in both normal physiological processes and the development of systemic pathologies, including cancer [2]. EVs, corresponding to exosomes and microvesicles, are secreted by the vast majority of cells and tissues and can be found in almost all body fluids. Bioactive signaling molecules, such as proteins, lipids, and nucleic acids, are selectively and precisely incorporated into EVs through the controlled mechanisms of cargo loading [3,4]. Individual cells are capable of secreting EVs of different molecular compositions. Moreover, the molecular cargo of EVs secreted by the same cells can vary depending on the state of the cell and numerous external stimuli [5]. Small EVs, commonly referred to as exosomes or exosome-like vesicles, are actively secreted by cancer cells and significantly contribute to carcinogenesis, including ovarian cancer. This includes primary tumor growth, microenvironmental remodeling, invasion, metastasis, and multidrug resistance [6,7,8,9].

Liquid biopsy is becoming a powerful tool for cancer diagnosis and surveillance due to its minimal invasiveness and high accessibility. Given that tumor cells produce EVs in significantly higher amounts than normal cells, it is not surprising that these structures are of steadily increasing interest in the context of cancer diagnosis. Molecules within EVs have several advantages over tissue and serologic markers as well as circulating cell-free nucleic acids as potential cancer markers [10,11]. These advantages include the high stability of the molecules due to their surrounding by the lipid bilayer of the membrane, high informativeness and representativeness due to the large diversity of EVs present in body fluids produced by all cells involved in tumorigenesis, and selectivity due to the aforementioned directed and precise cargo selection. It is also important to note that EVs have a higher concentration of molecules compared to cell-free circulating molecules [12,13]. Currently, a significant amount of data has been accumulated on tumor-dependent changes in the molecular composition of EVs. Many potential diagnostic, prognostic, and predictive markers, including single EV molecules or various combinations, have been proposed [11]. In particular, numerous studies have identified prospective molecular markers for OC diagnosis, including EV-miRNA panels [14,15,16,17,18]. At the same time, the sets of miRNAs proposed in different studies differ greatly and sometimes practically do not overlap [19]. The apparent contradictions can be attributed to several factors, including the significant natural heterogeneity of EVs and variations in their composition, as well as the diversity of methods and technologies used to isolate and analyze EVs’ molecular composition [20,21].

A major limitation of EV cancer marker screening is the complexity of distinguishing between normal and cancer-cell-derived EVs and the inability to selectively isolate tumor-derived EVs from the body fluids of cancer patients. This issue is especially relevant since most studies use blood (plasma or serum) as a source of EVs. It is important to note that the share of tumor-associated EVs in the total pool of blood vesicles is very small (about 1% according to different estimates) since the main producers of EVs are blood cells, immune cells, as well as endothelial cells and epithelial cells of various histological types [22]. It has been noted that biofluids in close proximity to a tumor may contain a higher concentration of tumor-derived DNA [23].

Previously, we suggested that local tissue-specific (or organ-specific) body fluids may be a more preferable source of EVs for tumor marker screening since they are located in the tumor growth area (washing the tumor growth zone) and are expected to be enriched in tumor-derived EVs. If such body fluids are also present in healthy individuals, they comprise a good alternative EV source for studying cancer-associated changes in EV composition and screening for diagnostic markers. Based on our previously published data, body fluids such as gastric juice (in the context of gastric cancer markers) and aspirates from the uterine cavity (for screening for gynecologic cancer markers) can be used [24,25]. In particular, our pilot study of miRNA composition in EVs from uterine aspirates (UAs) revealed a number of molecules that are differentially expressed in samples from OC patients compared to healthy individuals.

The deep sequencing (small RNA-seq) data presented here, obtained using an expanded UA sampling, confirmed significant differences in miRNA expression profiles in EVs from OC patients and non-cancer individuals. To elucidate the relationship of the identified panel of miRNAs with OC pathogenesis, we compared miRNA expression profiles in UA EVs with those obtained from the ascitic fluid of OC patients (AF EVs) as well as from the conditioned medium of cells isolated from OC ascites and transferred into the primary culture (ascitic cells, AC EVs).

We found that the vast majority of miRNAs upregulated or downregulated in UA EVs from OC patients compared to the control group (UA EVs from non-cancer patients, UA-N EVs) showed similar changes (i.e., upregulation or downregulation, respectively) when the AF EVs group was compared to the same control. Moreover, the levels of certain miRNAs were found to be co-modified (upregulated or downregulated, respectively) in AF EVs even when compared to UA EVs from OC patients (UA-OC EVs). More importantly, although the miRNA profiles in ascites EVs and AC EVs were generally very different, we observed the common changes in miRNA composition in AF EVs and AC EVs compared to UA-N EVs. Moreover, the set of miRNAs that were significantly altered in AF EVs compared to control UA EVs is almost identical to the set of miRNAs that were differentially expressed in AC EVs compared to the same control group (40 out of 44). In other words, we identified a set of miRNAs that are coordinately altered in EVs from OC-related body fluids, including uterine aspirates from OC patients, ascitic fluid from OC patients, and conditioned media from primary-cultured OC cells. This approach confirmed that most of the changes in miRNA levels in EVs from uterine aspirates of OC patients compared to healthy donors (differentially expressed miRNAs) are indeed related to the pathogenesis of OC. The results of this study support the prospective use of EVs from local body fluids as a source of cancer screening markers in general and the use of UA EVs to search for diagnostic markers of OC in particular. The OC-associated miRNAs identified in this study can be considered as promising diagnostic markers. The study highlights the essential function of EV miRNAs in OC and provides new insights into the clinical applicability of EV miRNA-based liquid biopsies.

## 2. Materials and Methods

### 2.1. Clinical Samples

Clinical materials were collected at the Oncogynecology Department of the N.N. Blokhin National Medical Research Center of Oncology prior to surgical or other types of treatment. Uterine aspirate samples were collected using a type C Pipelle probe from patients with a histologically verified diagnosis of ovarian cancer (UA-OC, N = 56, mean age 52.3 years (SD 9.3)) and donors with no history of cancer (control group, UA-N, N = 25, mean age 47.8 years (SD 8.3)). The tumor clinical and morphological characteristics were determined according to the FIGO classification (Table 1).

The initial volume of the UAs ranged from 0.5 to 1.5 mL. The samples were then diluted in 5 mL of ice-cold PBS immediately following collection. A total of 58 samples of ascitic fluid (AF) were obtained under sterile conditions from the patients with verified OC. All AF samples were obtained from patients diagnosed with OC high-grade adenocarcinoma. Sample volumes varied from 5 to 500 mL, with 10 to 25 mL samples being the most frequently collected. Sampling was performed before surgery or other treatment. The study protocol was approved by the Ethics Committee of the N.N. Blokhin National Medical Research Center of Oncology (Ethics Committee Permission Protocol for Project № 22-15-00375 dated 10/06/2022; Approval Protocol №5). All the experiments were conducted in concordance with the principles of the Declaration of Helsinki. Written informed consent was sought and obtained from all participants.

### 2.2. Sample Processing

Each sample was processed within one hour after collection and kept on ice throughout the processing. Under sterile conditions, the ascitic fluid samples were transferred into centrifuge tubes and centrifuged at 300× *g* for 20 min (4 °C). The precipitate, consisting mainly of ovarian tumor cells, was used to establish short-term primary cultures. The obtained supernatant (after 5-fold dilution with ice-cold PBS) was further purified using a protocol similar to that used for uterine aspirates, as described previously [24]. After 30 s of vortexing, UA samples were subjected to the following protocol of sequential centrifugation (4 °C): 800× *g* (20 min), 2000× *g* (30 min), and 10,000× *g* (30 min), allowing for the removal of cells, cell debris, and large vesicles, respectively. Next, the final supernatant was frozen at −80 °C until small EV isolation.

### 2.3. Primary Culture of Ascites Cells and Collection of Conditioned Medium

Cells were transferred into primary culture in accordance with the protocols described [26], with some modifications. In brief, the pellet obtained from the first round of centrifugation of malignant ascites was dissolved in sterile DMEM High Glucose (4.5 g/L) (#C420п, PanEco, Moscow, Russia) and supplemented with 10% fetal bovine serum (FBS) (#SV30160.03, HyClone, Pasching, Austria), 100 U/mL penicillin, and 100 μg/mL streptomycin (#A065, PanEco), then transferred to a cell culture plate (#20101, SPL Life Sciences, Pocheon, Republic of Korea) and maintained at 37 °C in a 5% CO_2_ environment. The next day, fresh medium was used to wash off any remaining blood cells. The cells were then grown to form a monolayer and then seeded into five 60 cm^2^-culture dishes following three passages. The next day, the medium was changed to an exosome-depleted medium and the cells were grown to a subconfluent monolayer. The conditioned medium was collected and subjected to sequential centrifugation, as described in step 2.2, except that the supernatant was not frozen but stored at 4 °C for a maximum of 1 week. The procedure was repeated twice, and the resulting purified medium was pooled to isolate EVs. The exosome-depleted medium was obtained by using DMEM-diluted (4.5-fold) FBS that had been pre-cleared of native vesicles through ultracentrifugation at 110,000× *g* overnight (sterile, 4 °C).

### 2.4. Cell Staining

Primary-cultured cells were seeded into the glass chambers for 48 h under standard cultivation conditions. Then, cells were washed with warm PBS solution and fixed with 3,7% PFA (#158127, Sigma−Aldrich, St. Louis, MO, USA) dissolved in PBS for 10 min. Cells were washed by PBS twice and additionally treated with 0.1% Triton X-100 (#T8787, Sigma−Aldrich) in PBS for 5 min. Washed cells were stained with primary anti-vimentin and pan-cytokeratin mouse monoclonal antibodies. Nuclei were additionally stained with DAPI (#D9542, Sigma−Aldrich). ProLong™ Gold Antifade Mountant was applied (#P10144, Invitrogen, Waltham, MA, USA). A Carl Zeiss Axioplan-2 fluorescent microscope and an AxioCam Hr (Carl Zeiss, Moscow, Russia) with AxioVision software (Carl Zeiss, Rel.4.6) were used to obtain the final images. The following antibodies were used: anti-pan cytokeratin [AE1/AE3 + 5D3] (ab86734, 1:100, Abcam, Cambridge, UK) and anti-vimentin [clone V9] (#M072501, 1:100, Agilent DAKO, Santa Clara, CA, USA). Alexa-546-labeled goat anti-mouse IgG (H + L) highly cross-adsorbed secondary antibodies (#A11030, Invitrogen) were used as secondary antibodies.

### 2.5. Extracellular Vesicles Isolation

EVs were isolated from three biological fluids: uterine aspirates, ascitic fluid, and conditioned medium (CM). Section 2.2 describes the preprocessing steps for sample purification. To isolate small EVs, a standard protocol of differential ultracentrifugation, as described by Théry et al. [27], was employed with minor modifications in accordance with the protocol previously described [24,28].

### 2.6. Nanoparticle Tracking Analysis (NTA)

The concentration and size distribution of extracellular vesicles were quantified in all samples using the NanoSight LM10 HS nanoparticle tracking analysis instrument, following the previously published protocol [24]. The non-parametric Wilcoxon rank-sum test was used to test the significance of differences (*p* < 0.05) in size and EV concentration between UA-OC EVs and UA-N EVs.

### 2.7. EV Visualization

The EVs were analyzed using transmission electron microscopy (TEM) with negative staining and cryo-electron microscopy (cryo-EM). The electron microscope JEM-1400 (JEOL, Akishima, Japan) was utilized for the generation of TEM images in accordance with the methodology delineated by Skryabin et al. [24], which were subsequently processed through the utilization of ScanEV software [29].

For the cryo-EM investigation, 3 µL of sample suspension was applied to the perforated lacey carbon support film (EMCN, Beijing, China). The support film was pre-treated with glow discharge with an EasyGlow device (TedPella, Redding, CA, USA). The grids were vitrified in liquid ethane with EM GP2 cryoplunger (Leica Microsystems, Wetzlar, Germany) with a blot time of 12 s. The images were acquired with 40e/A2 and −2 μm defocus using a JEM-2100 transmission electron microscope (JEOL, Tokyo, Japan) equipped with a DE-20 camera (Direct Electron, San Diego, CA, USA) and automated with SerialEM software ver.3.8 [30]. The obtained images were processed manually using ImageJ software ver. 1.54d [31].

### 2.8. Immunoblotting and Antibodies

The total protein concentration in EV samples and cells lysed in RIPA buffer (50 mM Tris-HCl, pH 7.5; 150 mM NaCl; 1 mM EDTA; 0.5% sodium deoxycholate; 1% Triton X-100 and 0.1% SDS) supplemented with an EDTA-free cOmplete protease inhibitor cocktail (# 05892791001, Roche Diagnostics, Mannheim, Germany) was determined using Bradford reagent (#500-0006, Bio-Rad Laboratories, Munich, Germany), following the manufacturer’s guidelines, and measured with a Benchmark Plus microplate spectrophotometer (Bio-Rad Laboratories). A total of 10 µg of protein were mixed with 4× sample buffer (0.2 M Tris-HCl (pH 6.8), 2.8% β-mercaptoethanol, 40% glycerol, 4% SDS, and 0.1% bromophenol blue) and then boiled at 95 °C for 10 min. The samples were subjected to 15% SDS-PAGE, transferred onto a PVDF membrane (Merk Millipore Ltd., Carrigtwohill, Ireland), and blocked with 5% bovine serum albumin (#0332-100G, VWR Life Science, Radnor, PA, USA) in Tris-buffered saline (TBS) with 0.1% TWEEN-20 for 1 h at room temperature (RT). The membranes were incubated overnight at 4 °C with primary antibodies, washed three times with TBS/TWEEN-20, and incubated with secondary antibodies for 1 h at RT. The protein bands were visualized using a SuperSignal™ West Femto Maximum Sensitivity Substrate (#34095, ThermoFisher Scientific, Rockford, IL USA). The resulting images were captured with a Kodak GelLogic 2200 Imaging System. The following antibodies were used: anti-Flotillin-2 (#3436S, 1:1000; Cell Signaling Technology, Danvers, MA, USA), anti-CD9 (#13174, 1:2000; Cell Signaling Technology), anti-TSG101 (ab125011, 1:5000; Abcam, UK), anti-Stomatin (#sc-134554, 1:500; Santa Cruz Biotechnology, Dallas, TX, USA), anti-PCNA (#sc-7907, 1:500; Santa Cruz Biotechnology), anti-mouse horseradish peroxidase-conjugated goat polyclonal antibodies (#2367, 1:5000; Cell Signaling Technology) and anti-rabbit horseradish peroxidase-conjugated goat polyclonal antibodies (#29902, 1:80,000; Cell Signaling Technology).

### 2.9. RNA Extraction and Small RNA Deep Sequencing

EV RNA isolation and analysis was conducted using a previously described protocol [24], with qualitative and quantitative analysis performed on an Agilent 2100 Bioanalyzer for small RNA profiles (Agilent Technologies, Santa Clara, CA, USA) and sequencing conducted on a HiSeq1500 (Illumina, San Diego, CA, USA) instrument.

### 2.10. Small RNA-Seq Data Analysis

The obtained sequence results in FASTA files were analyzed as described previously [24]. Differential expression (DE) analysis was carried out using the standard pipelines of three software packages, namely edgeR (version Galaxy 3.36.0) [32], Limma-voom (version 3.50.1) [33], and DESeq2 (version 2.11.40.8) [34]. The Benjamini–Hochberg method was used to control the expected proportion of false hypothesis rejections (FDR), and the obtained value was used to assess the significance of the result. Adjusted *p*-values less than 0.01 were considered statistically significant. The selection of DE miRNAs was based on the criteria FDR < 0.01, logCPM > 2, and fold change > 2 estimated by each package. The small RNA-seq data for this study have been deposited in the NCBI Sequence Read Archive (SRA) under project number PRJNA1070217 (https://www.ncbi.nlm.nih.gov/sra/PRJNA1070217).

### 2.11. Gene Ontology Enrichment Analysis

The miRTarBase v.9.0 database (https://miRTarBase.cuhk.edu.cn/, accessed on 27 October 2023) was used to search for target genes regulated by DE miRNAs labeled as ‘Set 2’. In order to ensure the reliability of the miRNA−target interactions, we used the following criteria: each interaction had to be validated by at least three methods, including the reporter assay, Western blot, and qPCR. The 217 genes obtained were analyzed for functional annotations using the Database for Annotation, Visualization, and Integrated Discovery (DAVID; v.6.8, https://david.ncifcrf.gov/, accessed on 1 November 2023). Gene sets with a false discovery rate (FDR) ≤ 0.05 obtained by the Benjamini−Hochberg method were considered. Enrichment analysis was performed on all proteins, extracting the top 20 functional GO terms of Biological Process, Cellular Compounds and Molecular Functions.

### 2.12. Statistics

Statistical tests for certain assays are described in their respective subsections. Based on the NTA-measured particle size and concentration, values of mean, mode, percentile data (10th and 90th), standard deviation, and confidence interval were calculated using Wolfram Mathematica ver. 11 (Wolfram Research, Champaign, IL, USA) software. A significant Kruskal–Wallis test followed up by a Dunn’s multiple comparisons test was used to compare the NTA-measured data of EVs isolated from aspirates, ascites, and the CM of ascitic cells. For statistical analysis, we used the statistical software package GraphPad Prism ver. 8.0.0 package for MS Windows, engineering-mathematical package Wolfram Mathematica ver. 11.

## 3. Results

The study design included an analysis of the EV small RNA transcriptome from the uterine aspirates (UAs) of OC patients (UA-OC) and healthy donors (UA-N), the ascitic fluid of OC patients (AF), and the conditioned cell medium of tumor cells isolated from malignant ascites and grown in primary cultures (AC). The scheme of the study is shown in Figure 1.

### 3.1. Differential Expression of miRNAs in EVs Isolated from the Uterine Aspirates of Ovarian Cancer and Non-Cancer Patients

Previously, using a pilot sample, we showed significant differences in miRNA profiles in EVs isolated from the uterine aspirates of ovarian cancer patients and healthy donors [24]. The revealed changes in miRNA composition of UA EVs indicate that differentially expressed (DE) miRNAs could be promising markers of ovarian cancer.

To test this hypothesis, here we used an expanded sampling of UAs (N = 81), including 56 UAs from epithelial ovarian cancer patients and 25 UAs from donors without a history of cancer. Data on clinical and morphological features of the included OC patients, specifically the histologic type of tumor, grade and stage of disease (according to FIGO), are presented in Table 1. EVs from all the UA samples were isolated by ultracentrifugation, as previously described [24,27], and characterized according to the ISEV guidelines [35] using three independent methods: TEM, cryo-EM, NTA, and Western blot analysis.

The size and morphology assessed by TEM showed the presence of a large number of membrane-coated particles of cup-shaped morphology, typical for this type of assay (Figure 2a). The average EV size distribution and concentration in the obtained preparations were analyzed by NTA. The mean size of the particles varied from 104 to 183 nm, with a mode of 67 to 141 nm in different individual EV preparations. The mean size of EVs and the median across the entire sample range were 139 nm (SEM 2.3) and 123 nm (SEM 2.4), respectively. The concentrations of EVs ranged from 10^11^ to 10^13^ particles per mL, averaging 6.61 × 10^12^ particles/mL for the entire sample. The mean size and concentration of EVs in preparations obtained from OC patients (UA-OC EVs) and individuals with no history of cancer (healthy donors (UA-N EVs)) had no significant differences (Mann–Whitney U test, *p* > 0.05). The NTA data on the EV diameter distribution of a population of UA-OC and UA-N samples, as well as the average size characteristics, are shown in Figure 2b,c. In addition, when analyzing the TEM images, we noticed some vesicles with atypical morphology similar to those that have been previously found by cryo-electron microscopy of EVs from several other types of body fluids [36,37,38]. These morphological types include double- and multi-layered vesicles, as well as elongated and dumbbell-shaped or budding-like vesicles. Given that a comprehensive analysis of the morphology of EVs derived from uterine aspirates has not been conducted previously, it was of interest to ascertain the prevalence of vesicles exhibiting atypical morphology and their relative abundance. To this end, a more detailed characterization of EV morphotypes was performed for a single UA-EV sample using cryo-electron microscopy, which preserves the native structure of the vesicles. The analysis of 300 cryo-EM images yielded the identification of several major morphological categories of EVs, distinguished based on their shape, number of surrounding lipid membranes, the presence of internal vesicles, and electron density. However, it should be noted that there may be more morphological variants, given the vagueness of the classification criteria and the presence of different combinations. Examples of images corresponding to these morphological types and their proportions are shown in Figure 2d,e.

Next, the vesicular nature of the particles in the obtained preparations was further confirmed by the high level of exosomal markers. In accordance with ISEV recommendations, the levels of several proteins from different intracellular compartments belonging to different functional types were analyzed in each EV sample (namely, tetraspanin CD9, the ESCRT complex component TSG101, and the lipid microdomain protein flotillin-2). We also analyzed the level of stomatin, a lipid raft protein from the same SPFH family, which we have previously proposed as a highly specific exosomal marker (Figure 2f) [28]. The absence of cellular proteins of non-vesicular origin was confirmed by PCNA nuclear protein analysis (negative control). Lysates of OC cells isolated from ascites and grown in primary culture were used as a positive control for PCNA expression. The analyzed markers, including stomatin, were detected in all samples, and, with the exception of flotillin-2, all showed higher levels in UA EVs than in cells (EV enrichment phenomenon characteristic of EV markers). The level of flotillin-2 was higher in cells than in EVs, but it should be emphasized that, in our case, the control was not parental EV-producing cells but highly malignant cells isolated from ascites at advanced stages of OC. Since flotillin-2 is upregulated in highly malignant cells according to the literature and our previously published data [28], the level of flotillin in primary-cultured ascitic cells may exceed that in EVs from uterine aspirates.

RNA isolation from EVs was performed using the method previously shown to be the most efficient in terms of RNA concentration and enrichment of small RNAs, including miRNAs [39]. After confirmation of the quality of the preparations and analysis of the small RNA size distribution using an Agilent 2100 Bioanalyzer (example analysis in Figure 3a), transcriptome was profiled by NGS small RNA-seq. The results of the bioinformatic analysis expectedly showed the presence of a wide range of noncoding RNAs, including piwiRNA, miRNA, snRNA, snoRNA, vaultRNA, etc., along with short fragments originating from protein coding and structural RNAs, such as rRNA, mRNA, lncRNAs, as well as various pseudogenes and intergenic repeats. The prevalence of different RNA classes in the obtained EV samples are presented in Figure 3b. Notably, piwiRNAs appeared to be the most represented class of all the small RNAs in EVs. The proportion of miRNAs was 25.7% on average. No differences in the percentage of miRNAs and piwiRNAs were found between the UA-OC and UA-N EV groups.

An initial comparison of EV miRNA content in UA-OC and UA-N using the EdgeR software package showed that the miRNA profiles have pronounced differences and that the samples clustered well according to the multidimensional scaling (MDS) plot (Figure 3c). To identify maximally expressed changes, we further used three different software packages instead of one, including EdgeR ver.3.36.0, Limma-voom ver.3.50.1 and DESeq2 ver.2.11.40.8, to search for differentially expressed miRNA molecules (DE miRNAs) in OC samples and control groups. The data were compared by a DEApp interactive web interface for differential expression [40]. Differences in meeting the criteria of FDR < 0.01 and FC > 2 were considered reliable only if the reported values were obtained independently by each software package. The comparison revealed 79 DE miRNAs (hereafter referred to as ‘Set 1’), of which 34 were upregulated and 45 were downregulated in the OC group compared to normal controls (Appendix A). Importantly, the use of three software packages significantly affected the results of the DE estimation. Specifically, when using *EdgeR* alone, statistically significant differences were observed for 133 miRNAs.

Within the group of UA-OC samples, miRNA profiles were then compared according to clinical and morphological characteristics of the tumors, such as histological type, extent, and stage of disease. The analysis revealed significant differences in miR-3180-3p and miR-383-5p levels between the grade 1 and grade 2/3 groups. Specifically, both miRNAs were reduced in the high-grade (G2/G3) group (N = 45) compared to the low-grade (G1) group (N = 9) (Appendix A). No significant differences in miRNA expression were found in the other comparison groups.

To identify the miRNA alterations most relevant to ovarian cancer, we used two approaches (see Figure 1). The first approach was to narrow the corridor of statistical significance for the differential expression (DE) score. Differences were considered reliable if the FDR value was <0.001 and only if it was obtained independently by each of the three software packages. This approach reduced the number of DE miRNAs to 35, of which 15 were upregulated and 20 were downregulated in UA-OC EVs compared to controls (hereafter ‘Set 2’, Table 2).

GO enrichment analysis revealed a strong association of DE miRNAs from ‘Set 2’ with key processes involved in tumorigenesis, including proliferation, angiogenesis, negative regulation of apoptosis, positive regulation of migration activity, etc., as well as key cancer-associated signaling pathways, including upregulation of the MAP kinase cascade. The top 20 processes according to FDR values in the categories “Biological Processes” (BP), “Molecular Functions” (MF), and “Cellular Compounds” (CC) are listed in Figure 3d.

The second approach was to determine whether the detected changes in miRNA profiles were indeed attributable to OC and to identify miRNA molecules most closely associated with OC pathogenesis (see Figure 1). For this purpose, we compared the spectrum of DE miRNAs in UA EVs (‘Set 1’) with the spectra of miRNAs differentially expressed in EVs isolated from biological fluids directly related to OC, namely ascitic fluid (AF EVs, ascitic fluid-derived EVs) and a conditioned medium of OC cells isolated from ascites and grown in primary cultures (AC EVs, ascitic cells derived EVs). We hypothesized that, if the identified miRNAs are indeed involved in OC progression, corresponding changes in their levels would also be observed in AF EVs and AC EVs. In such a scenario, the sets of DE miRNAs should overlap when comparing UA-OC, AF, and AC EVs with UA-N EVs, and at least a significant proportion of the miRNAs should display coordinated DE (or differential co-expression). For example, miRNAs significantly upregulated in UA-OC EVs should also be significantly upregulated in AF EVs and AC EVs compared to the same controls.

### 3.2. EVs from Malignant Ascites and the Uterine Aspirates of OC Patients Show a Common Spectrum of miRNA Alterations Compared to Non-Cancer Individuals (Differential Co-Expression)

To identify miRNAs differentially expressed in EVs from the **a**scitic **f**luid of OC patients (AF EVs) compared to the matched control group (UA-N EVs), we used ascites specimens from patients with high-grade serous adenocarcinoma. Thus, the AF group (N = 58) was fairly homogeneous in terms of histologic type and disease stage and represented the most aggressive type of OC. Isolation and storage of EVs, as well as RNA isolation, were performed under the same conditions and following the same procedures as for UA, which is essential to ensure the reliability of the subsequent comparison of EV-miRNA composition. The approach described above was used to characterize EVs, with the exception of cryo-EM. Data supporting the vesicular nature of the particles, including the average size distribution and morphology, as well as the levels of exosomal markers, are summarized in Figure 4a–c.

The mean size of particles assessed by NTA varied from 112 to 186 nm, with a mode of 66 to 126 nm in different individual EV preparations. The mean size and median size of EVs across the entire sample were 143.6 nm (SEM 4.7) and 122.9 nm (SEM 4.5), respectively (Figure 4c). The concentration varied from 10^11^ to 10^13^ particles per mL in individual preparations, and the average EV concentration across the entire sample was 3.43 × 10^12^ particles/mL. For convenience, Appendix A summarizes the NTA data on EVs isolated from uterine aspirates, ascites, and a conditioned medium of cultured cells. The results of TEM analysis showed the presence of vesicles with the atypical morphology described above along with classically shaped EVs. However, they were not analyzed in detail by cryo-electron microscopy, and their percentage was not determined.

RNA isolation from EVs and small RNA-seq transcriptome profiling followed a similar approach to UA EVs. Figure 4d shows the size distribution of small RNAs. The pie charts in Figure 4e indicate the percentage of different small RNA classes present in AF EVs. Similar to UA EVs, piwiRNAs were the most abundant in AF EVs, accounting for nearly 60% of the small RNA transcriptome. MiRNAs were the second most abundant in AF EVs averaging 21.8% of the small RNA transcriptome. Next, we compared the miRNA expression profiles in AF EVs with those in UA-OC EVs and UA-N EVs. The MDS plots in Figure 4f show well-defined clusters, indicating the dissimilarity of sample distributions between the AF and UA-N groups, and less pronounced but still distinct differences when comparing the AF to the UA-OC group (Figure 4g).

When comparing miRNA levels in AF EVs and UA-N EVs, the same strict approach was applied, i.e., the search for DE miRNAs was performed using three software packages and the differences were considered significant only if FDR < 0.01 and FC > 2 criteria were detected by each program. A total of 281 DE miRNAs were identified, of which 134 were upregulated and 147 were downregulated in AF EVs. This abundance of DE-miRNAs was quite expected, given that EVs had been derived from different body fluids. However, our attention was focused on the presence of the ‘Set 1’ molecules among them, i.e., miRNAs differentially expressed in UA-OC EVs (Appendix A).

Therefore, we compared the ‘Set 1’ miRNAs with the panel of DE miRNAs obtained by comparing AF with UA-N EVs. We found a strong overlap between these sets, with a significant number of miRNAs showing changes in the same direction (increase or decrease, respectively). Specifically, out of the 79 miRNAs that were differentially expressed in UA-OC EVs compared to UA-N EVs (‘Set 1’, taken as 100%), 60 miRNAs (76%, referred to as ‘Set 3’) exhibited similar co-directed changes (differential co-expression) in the AF EVs compared to the same control (Figure 5a; Appendix A). In both comparisons, 23 miRNAs were upregulated and 37 miRNAs were downregulated. These subsets are labeled as ‘Subset 3ab’ and ‘Subset 3cd’, respectively (Figure 5b). The remaining 24% (19 miRNAs, 3e) showed no significant changes in the AF EV group. Remarkably, we did not find any miRNAs that significantly changed in the opposite direction!

Even more impressive was the finding that, among the 60 ‘Set 3’ miRNAs, 29 miRNAs (i.e., almost half) additionally showed similar (i.e., significant and co-directed) changes even when comparing AF EVs with UA-OC EVs, of which 9 miRNAs were upregulated (Subset “3a”) and 20 were downregulated (Subset “3c”) (Figure 5b,c; Appendix A).

The levels of the remaining 31 of 60 miRNAs did not differ significantly between the indicated groups, and only one miRNA, miR-27a, was altered in the opposite direction, being decreased in AF EVs compared to UA-OC EVs, although it was increased in both of these groups compared to the control.

The obtained results suggest that most of the EV-miRNAs, whose levels are significantly altered in the uterine aspirates from OC patients, are similarly up/downregulated in EVs from ascitic fluid, thus their dysregulation is likely to be involved in the development of ovarian cancer. To further confirm this hypothesis, we analyzed the content of identified DE miRNAs in EVs secreted by ovarian cancer cells in primary culture.

### 3.3. MiRNAs Differentially Co-Expressed in EVs from Primary-Cultured OC Cells, Malignant Ascites, and the Uterine Aspirates from OC Patients Compared to Non-Cancer Subjects

Despite the theoretically expected enrichment of AF EVs with tumor-associated EVs, it is clear that ascitic fluid should contain a certain proportion of vesicles of a non-tumor origin. Therefore, it was of interest to analyze the levels of the identified DE-miRNAs in EVs that would have been produced solely by OC cells. For this purpose, cells isolated from OC malignant ascites were transferred into primary culture. In total, primary cultures were obtained from 23 ascites specimens, including 7 specimens from which AF EVs had been previously isolated and 16 independently collected specimens. All specimens were obtained from patients diagnosed with high-grade serous adenocarcinoma.

Morphologic analysis of cultured cells and their immunofluorescent staining were performed after three passages. The epithelial origin of the cells and the epithelial-mesenchymal transition were confirmed by immunofluorescence staining for pan cytokeratin and vimentin, respectively (Figure 6a). Isolation and validation of vesicles secreted by OC cells in vitro (**A**scitic **C**ells-derived EVs, AC EVs) was performed similarly according to ISEV guidelines.

TEM analysis revealed the presence of a large number of vesicles with a typical size and cup-shaped morphology (Figure 6b). Similar to the vesicles found in the aforementioned biofluids, the AC EVs samples also contained a small number of vesicles with atypical morphology, such as dumbbell-shaped, elongated, double and multilayered vesicles, as well as non-membrane particles.

According to the NTA data, the mean size of the particles varied from 110 to 144 nm, with a mode of 68 to 151 nm in different single preparations. The mean and median size of EVs over the entire sample were 128 nm (SEM 2.5) and 112 nm (SEM 2.8), respectively (Figure 6c). The concentration varied from 10^11^ to 10^13^ particles per mL in individual preparations, and the average EV concentration over the entire sample was 3.91 × 10^12^ particles/mL. The Kruskal–Wallis test indicated a significant difference in the mean size of EVs isolated from the aspirates, ascites, and CM of ascitic cells (*p* < 0.05). A subsequent Dunn’s multiple comparisons test revealed that AC-EVs were larger than EVs derived from both aspirates and malignant ascites (adj. *p* < 0.05).

All EV preparations were characterized by a high content of exosomal markers and their enrichment compared to parental ascitic cells, especially pronounced for stomatin and CD9 proteins (Figure 6d). RNA isolation from AC EVs, purification, and size spectrum analysis were performed, as described above (Figure 6e).

NGS small RNA-seq data showed that AC EVs, UA EVs, and AF EVs had a similar distribution of RNA classes. In AC EVs, piwiRNAs accounted for the largest proportion of the small RNA transcriptome, with an average of 61.5%. MiRNAs accounted for an average of 13.9% (Figure 6f). Comparison of miRNA profiles revealed maximally pronounced differences between AC EVs and the control group of UA-N EVs, as shown in the MDS plot in Figure 6g. The miRNA profiles in AC EVs were also substantially different from those in ascitic fluid EVs, although the difference was less pronounced (Figure 6h). Differential expression analysis using the same three software packages and inclusion criteria (FDR < 0.01, FC > 2, logCPM > 2) identified 264 miRNAs with significantly altered levels in AC EVs compared to control.

As with AF EVs, we focused on those miRNAs that were found to be differentially expressed in EVs from the uterine aspirates of OC patients. Therefore, we compared the obtained list of DE miRNAs in AC EVs with ‘Set 1’ and found that 44 miRNAs (56%) were differentially co-expressed, that is, significantly changed in the same direction in both comparisons (labeled as ‘Set 4’, Figure 7a; Appendix A). Among them, 16 miRNAs were upregulated in both UA EVs and AC EVs (Subset “4ab”), and 28 were downregulated (Subset “4cd”). A total of 31 miRNAs (39.2% from ‘Set 1’) showed no DE in AC EVs compared to the control (Figure 7b). Notably, only three DE miRNAs (3.8%) were significantly altered in the opposite direction. The comparison of ‘Set 4’ and ‘Set 3’, i.e., the sets of DE miRNAs in EVs extracted from an ascitic cell culture medium and EVs isolated from ascitic fluid, yielded even more impressive results—these sets not only overlapped, but were almost identical (specifically, 40 out of 44 miRNAs from ‘Set 4’ were present in ‘Set 3’) (Figure 7c).

Given the high dissimilarity of miRNA profiles in AF EVs and AC EVs in general (see MDS plot, Figure 6h), this striking similarity of DE miRNA sets in EVs from these biofluids means that changes in these specific miRNAs (40 out of 60 DE miRNAs in ascites EVs) are driven by alterations in the miRNA composition of tumor-derived EVs, indicating that the identified changes are associated with ovarian cancer. As with AF EVs, we aimed to determine if any DE miRNAs from ‘Set 4’ were also differentially co-expressed compared to not only the control group but also to OC patient aspirates, i.e., those miRNAs that were simultaneously up- or downregulated in AC EVs compared to both UA-N EVs and UA-OC EVs. The comparison confirmed the presence of such molecules, among which three were upregulated (Subset “4a”) and 10 were downregulated (Subset “4b”) (Figure 7b,d). These molecules accounted for 29.5% of ‘Set 4’. Since most of the molecules from subsets “4a” and ”4b” are also present in subsets 3a and 3b, this means that their levels in body fluids change in the same direction (increase or decrease) as they become more relevant to OC. In other words, the levels of such miRNAs consistently increase or decrease in the series “UA-N EVs–UA-OC EVs–AF EVs–AC EVs”.

In summary, we identified 40 miRNAs that exhibited significant co-directed changes (FDR < 0.01, FC > 2) in EVs from all biofluids associated with ovarian cancer compared to the control group (labeled as ‘Set 5’). These biofluids include the uterine aspirates of OC patients, OC malignant ascites, and a conditioned medium of ovarian cancer cells isolated from malignant ascites and grown in primary culture.

Finally, we combined the results of the two approaches used in this study to identify miRNAs that are most strongly associated with OC pathogenesis. To achieve this, we compared the list of ‘Set 2’ miRNAs, which was derived from ‘Set 1’ by using the most stringent criteria for calculating DE (FDR < 0.001), with the lists of ‘Set 3’ and ‘Set 4’, corresponding to DE miRNAs that were simultaneously altered in the same direction in EVs from all OC-related biofluids compared to the control (UA-N EVs). We found that, out of 35 miRNAs (83%), 29 miRNAs were found to match in Sets 2, 3, and 4, indicating significant co-directed changes in both UA-OC and other OC-related biological fluids. Moreover, none of the miRNAs were significantly altered in the opposite direction.

Of these 29 molecules, 17 miRNAs exhibited co-directed changes in all three biological fluids. miRNAs that were upregulated included miR-27a-5p, miR-193a-5p, miR-5100, miR-625-3p, miR-125a-5p, miR-671-3p, miR-29b-1-5p and miR-23a-5p. Among the miRNAs that were downregulated were miR-451a, miR-376c-3p, miR-127-5p, miR-136-3p, miR-542-5p, miR-193b-3p, miR-99a-3p, miR-199a-5p and miR-130a-5p. The 12 remaining miRNAs exhibited comparable changes in at least the following two sets: miR-200b-5p, miR-92a-1-5p, miR-2110 and miR-484 were upregulated in UA-OC EVs and AF EVs; miR-4521, miR-136-5p, miR-495-3p, miR-337-3p, miR-98-3p, miR-152-3p, miR-655-3p and miR-487a-3p were downregulated in OC-related biofluids. Details are shown in Table 3.

These findings indicate the relevance of the identified EV-miRNA alterations in the pathogenesis of ovarian cancer and point to the high potential of the identified miRNAs as potential diagnostic markers.

## 4. Discussion

Ovarian cancer is one of the leading cancers in terms of incidence and mortality in women [41,42]. The average 5-year survival rate for patients with OC remains below 40%, despite some success with combined surgical and chemotherapeutic treatment [43,44]. OC is initially very responsive to platinum-based chemotherapy, and late diagnosis (mainly at stage III-IV) is the main reason for the high rate of recurrence and metastasis, which, in turn, is due to a lack of highly effective diagnostic markers that are both sensitive and specific.

Small extracellular vesicles, such as exosomes, are currently considered a promising source of liquid biopsy markers, in part, due to their potential for early detection of ovarian cancer. Molecules within these vesicles, particularly miRNAs, may be more suitable markers than cell-free circulating molecules that may result from non-specific cell death. In addition, several studies have shown that the concentration of miRNAs in exosomes is higher than that of free-circulating molecules in the same body fluids [12,45].

We previously hypothesized that uterine aspirates could serve as a valuable source of EVs for screening diagnostic markers of gynecologic cancers, such as OC. Indeed, it has been shown that OC cells are present in the uterus and that OC-dependent molecular alterations can be detected in aspirates [46,47,48]. At the same time, the molecular content of EVs from uterine aspirates has not yet been significantly studied. Our pilot study first revealed reliable differences in the miRNA expression profiles in EVs from the uterine aspirates of OC patients and healthy individuals, confirming the potential for further studies in this direction [24].

Here, we continued the study with an expanded sample of 81 specimens, including 56 UA-OC and 25 control specimens (UAs from non-cancer individuals). To select the most substantial miRNA changes, we searched for differential miRNA expression using three software packages: EdgeR ver.3.36.0, Limma-voom ver.3.50.1, and DESeq2 ver.2.11.40.8. Importantly, the results obtained using the three resources were significantly different from those obtained using only one. Specifically, 133 DE miRNAs were detected by using EdgeR alone, while, when three programs with the same selection criteria (FDR < 0.01, FC > 2) were used, the number of DE miRNAs was reduced to 79 (labeled ‘Set 1’, Appendix A).

When comparing the miRNA composition within UA-OC subgroups, statistically significant differences were found only according to malignancy grade. Specifically, two miRNAs, miR-3180-3p and miR-383-5p, were found to be downregulated in the G2-G3 group compared to the G1 group. Interestingly, miR-383-5p is well known for its anti-tumor activity, and its downregulated expression has been reported in many tumor types, including OC (and high-grade and clear cell OC in particular) [49,50]. Its effect is attributed to blocking OC cell proliferation and invasion.

Next, we narrowed down the number of DE miRNAs on one side by increasing the stringency of the selection criterion to FDR < 0.001. This resulted in a list of 35 miRNAs with the most significant expression changes (‘Set 2’, Table 2). On the other hand, we attempted to select the molecules most strongly associated with OC pathogenesis using an original approach. We hypothesized that, if the observed changes in miRNA profiles in UA-OC EVs indeed contribute to the progression of OC, then similar changes should be observed in EVs from biological fluids strongly related to OC. Therefore, we decided to look for miRNA changes in EVs present in malignant ascites from OC patients as well as in EVs produced by ascites cells grown in primary culture.

Ovarian cancer is commonly characterized by peritoneal dissemination of metastatic cells and the formation of malignant ascites [51]. Malignant ascites is found in more than one-third of OC patients and present in almost all the patients with recurrence. Ascitic fluid accumulates various cellular and soluble factors which are known to contribute to metastasis and chemoresistance [52]. Therefore, the miRNA content in AF EVs may serve as a valuable source for the investigation of EV molecules related to OC. Following the same reasoning, we added AC EVs that exclusively reflect the contribution of in vitro metastatic OC cells in carcinogenesis. To achieve this, cells were isolated from ascites and used to establish short-term primary cultures, and EVs were extracted from the conditioned medium of the cultured cells.

Isolation and analysis of EVs has been performed in accordance with ISEV recommendations [35]. In particular, the vesicular nature of the particles was confirmed by NTA, TEM, and exosomal marker analysis. Regarding the marker analysis, it is noteworthy that all types of EVs contained the stomatin protein, which we have previously proposed as a marker of small EVs [24,28]. Additionally, our data support that flotillin, often used in the literature as an exosomal marker, is not an ideal marker for EVs due to its high levels in tumor EVs. We had previously observed this phenomenon when comparing the flotillin content in EVs from different biological sources [28]. This is particularly important when comparing EVs from different biofluids, as the level of flotillin can vary significantly depending on the proportion of EVs secreted by tumor cells.

Importantly, all data for the compared EV groups were obtained under the same conditions, including EV isolation and storage procedures, RNA extraction and analysis methods, small RNA-seq protocols, etc. This ensures objectivity and eliminates potential sources of bias. All of these factors have a significant impact on the final outcome of the study, as supported by both the literature [53,54] and our data [39].

It should be noted that such an approach involves a comparison of rather heterogeneous samples where the “experimental” and “control” groups are represented by vesicles from different biological fluids. The question of whether such a comparison is correct is ambiguous. Comparisons of the molecular composition of EVs from different body fluids, such as EVs from ascites and blood plasma, can be found in the literature [55,56,57]. However, we believe that such an experimental design cannot be used as the sole approach in cancer marker screening because differences in the molecular cargo of EVs may be largely due to the diversity of EV origins. In other words, DE-miRNA panels obtained by comparing EVs from different biological sources cannot substitute the sets obtained by comparing EVs from the same type of body fluid (in our case, uterine aspirates). At the same time, the combination of these approaches provides extremely important information for the selection among DE-miRNAs of those molecules that actually have a functional relationship with the studied disease. In our case, the identification of miRNAs that are simultaneously up- or downregulated in EVs from the uterine aspirates of OC patients and EVs from biological fluids most closely associated with OC (such as OC ascites and a conditioned medium from ascites cells grown in primary culture) confirms their significance in the pathogenesis of OC.

Comparison of the DE miRNA panels in EVs from aspirates and ascites revealed that 60 of the 79 miRNAs (76%) up- or downregulated in UA-OC EVs were similarly up- or downregulated in AF EVs compared to controls. The remaining 24% of miRNAs showed no significant changes when comparing AF EVs to UA-N EVs. Importantly, no miRNAs were significantly dysregulated in the opposite direction. Moreover, almost half of these 60 miRNAs (29 miRNAs, 48%) showed co-directional expression changes when compared to UA-OC EVs. This means that the miRNAs that are up- or downregulated in UA-OC EVs, compared to controls, were even more up- or downregulated in AF EVs. All these data suggest that the identified changes are not random and are indeed associated with the development of OC.

When analyzing AC EVs, we found that, although the number of miRNAs showing co-directional changes (i.e., similar to those detected in UA-OC EVs) was lower, they still accounted for more than half of all differentially expressed molecules (44 out of 79 miRNAs). However, 3 out of 79 miRNAs showed significant changes in the opposite direction (2 miRNAs were upregulated in AC EVs but downregulated in UA-OC EVs; miR-375-3p showed the opposite pattern). It is important to note that, although AC EVs are strictly of an OC origin (unlike UA EV and AF EV preparations, where different proportions of vesicles must originate from different sources), their molecular content can be significantly altered by the transfer of OC cells into an in vitro culture. Indeed, the MDS plots show that the composition of EVs from conditioned media is maximally different from that of EVs from body fluids, including parental ascites (Figure 6g). This difference can be attributed to the change in growth conditions and the absence of the tumor microenvironment, which affects the miRNA composition of both the cells and the secreted EVs. Nevertheless, most of the miRNAs in AC EVs show co-directed changes when compared to controls and even to UA-OC EVs (13 out of 44 miRNAs).

Some of the most impressive and even astonishing results of the study were obtained when comparing the panels of miRNAs differentially co-expressed in EVs from OC-related biofluids compared to the control group (UA-N EVs) (‘Set 3’ and ‘Set 4’). It turned out that the set of DE miRNAs in EVs produced by ascites cells in primary culture (AC EVs) almost completely corresponded to the set of DE miRNAs in EVs from the ascitic fluid of OC patients (AF EVs). Thus, out of 44 miRNAs co-directionally dysregulated in UA-OC and AC EVs, 40 miRNAs (91%) were similarly co-directionally dysregulated in UA-OC and AF EVs (Figure 7c,d). Considering that the miRNA profiles of EVs from these biological fluids were generally highly divergent from each other, the observed similarity of DE miRNAs suggests a link between these miRNAs and OC, as well as their involvement in OC progression. Finally, we obtained three sets of miRNAs that appear to be the most promising in the studied context. The first one was obtained by comparing UA-OC and UA-N profiles using the most stringent criteria for the selection of DE molecules (FDR < 0.001)—it is represented by 35 miRNA molecules (‘Set 2’). The next set consists of 60 miRNAs co-differentially expressed in AF EVs and UA-OC EVs compared to UA-N EVs (‘Set 3’). Among them, 29 miRNAs show co-directed changes in the series “UA-N–UA-OC–AF” (Subsets “3a/3c”). The last one (‘Set 4’) is represented by 44 miRNAs that are co-expressed in AC EVs and UA-OC EVs compared to control UA-N EVs. Among them, 13 miRNAs show co-directed changes in the series “UA-N–UA-OC–AC” (Subsets “4a/4c”). When comparing these molecules, it is interesting to highlight the common miRNAs present in both subsets “3a/3c” and subsets “4a/4c”, i.e., those miRNAs whose differential co-expression becomes more pronounced as the relevance of the biological fluid to ovarian cancer increases (i.e., in the series “UA-N–UA-OC–AF–AC EVs”). These include the following ten miRNAs: miR-193a-5p, miR-3615, miR-455-3p, miR-199a-5p, miR-199a-3p, miR-199b-3p, miR-196b-5p, miR-196b-3p, miR-143-3p, miR-497-5p. Of particular interest is the set of miRNAs obtained by comparing DE miRNAs from ‘Set 3’ and ‘Set 4’ with ‘Set 2’ (Table 3).

Interestingly, among the DE miRNAs identified in our study, there were a number of molecules with known tumor promoter or tumor suppressor activities whose involvement in the carcinogenesis and progression of OC had been repeatedly demonstrated. Moreover, alterations of some of them were preferentially found in the composition of secreted EVs. Such molecules include members of the miR-200 family, including 200a, 200b, and 200c, whose upregulation in tissues and blood (both free-circulating and as part of EVs) had been implicated in OC development [17,58,59], progression, and recurrence [60,61,62]. According to a meta-analysis published in 2020, miR-200 appears to be one of the most promising markers for the diagnosis of OC [63]. More information about these and other miRNAs, including those identified in this study, are available in the reviews by Nguyen et al. and Yoshida et al. [62,64].

Other examples of DE miRNAs that we found in UA-OC EVs and EVs from other OC-related biological fluids include miR-205, whose increased level is also characteristic of OC progression [65,66,67], miR-193a, miR-484 and miR-27a, which regulate several cancer-associated signaling pathways [68,69,70,71,72] and have also been associated with OC progression [16,73,74] and chemotherapy resistance [75,76]. Notably, according to our data, the increase in miR-27a-5p levels was accompanied by an increase in miR-23a-5p in EVs from all OC-related biofluids and miR-24-3p in UA-OC EVs. These miRNAs belong to a single intergenic miRNA cluster, miR-23a/24-2/27a, which encodes a ~2159-nt pri-miRNA transcript and is located on chromosome 19p13.12. This cluster is often upregulated in tumor tissue and is considered as a potential cancer marker. More information about this cluster and its constituent miRNAs can be found in the following reviews [77,78,79]. In the context of OC carcinogenesis, in addition to the repeatedly shown changes in miRNA expression from this cluster, its location on chromosome 19 is important. Specifically, it is localized in an amplicon upstream of CA125 (or MUC16), a well-characterized OC biomarker, and downstream of the Notch3 oncogene, which is overexpressed by this cluster [80].

When discussing miRNAs that were found to be downregulated in EVs from OC samples, it is important to mention several molecules that have well-documented tumor suppressor activity. One such molecule is miR-451a, which, according to our data, is downregulated in EVs from all OC-related biofluids. Moreover, in EVs secreted by primary-cultured cells derived from OC ascites, the level of this miRNA is decreased not only compared to controls but even compared to EVs derived from parental ascites. MiR-451a is a well-known suppressor involved in suppressing EMT, angiogenesis, proliferation, migration and other properties of malignant cells, including OC cells, by regulating of multiple signaling pathways [81]. Another example is miR-199. Both members of this family, miR-199a and miR-199b, were found to be downregulated in EVs from all OC-related samples. MiR-199 regulates the expression of key signaling molecules involved in carcinogenesis, including members of the IKK/NF-κB and PTEN/AKT pathways [82,83], JAG1/Notch1 signaling [84], HIF-1α and HIF-2α [85], CD44 [86], E-cadherin [87], and others. The literature strongly supports the association between the suppression of these miRNAs and OC progression [87,88,89]. Mir-136 is currently of great interest because its dysregulation is characteristic of most types of neoplasia, the vast majority of which show a suppressor role of this miRNA in tumor progression [90,91]. Mir-136 regulates at least five key cancer-related signaling pathways, such as the MAP kinase cascade and the JNK pathway, suppressing cell proliferation, survival, migration, and invasion and stimulating apoptosis [92]. In addition, miR-136-5p has shown significant potential as a prognostic and diagnostic marker in human cancers and as an effective mediator in cancer chemotherapy [93]. In the context of our study, it is important to note that the identified panel of 17 downregulated miRNAs included both its forms, mir-136-5p and mir-136-3p. There is also ample evidence for the tumor suppressor activity of miR-152, which has been shown in several types of cancer, including ovarian cancer, as well as its role in the sensitivity of cells to cisplatin [94,95,96]. MiR-495-3p has also been identified as one of the key miRNAs in several cancer types and has been shown to be downregulated in a variety of solid tumors [97].

Other known suppressor molecules, such as miR-342-3p [98], miR-424-5p [99,100], miR-193b-3p [101], miR-99a-3p [102], miR-455-3p [103] and miR-126-5p [104,105,106], were also found to be downregulated in the EVs from all OC-related biofluids. Interestingly, some studies found DE miRNAs in combinations similar to those identified here. For instance, Y. Jiang et al. compared miRNA profiles in ascites-derived OC cells and primary OC tumors and found a combination of five miRNAs that were most significantly downregulated in ascites-derived spheroids compared with primary tumor tissues [104]. Overexpression of these miRNAs resulted in stimulation of apoptosis and inhibition of OC cell invasion. In our data, four of these five miRNAs, namely miR-199a-3p, miR-199b-3p, miR-199a-5p, and miR-126-3p, were also significantly downregulated in the EVs from all OC-related fluids. Another example is a well-known study by Iorio, who identified aberrantly expressed miRNAs in human ovarian cancer compared to normal ovaries [58]. The most significantly overexpressed miRNAs were miR-200a, miR-141, miR-200c, and miR-200b, whereas miR-199a, miR-140, miR-145, and miR-125b1 were among the most down-modulated miRNAs. The list of miRNAs identified as being differentially expressed in the cited study largely overlaps with our sets. More information about these miRNA combinations can be found in Alsharmrani’s review [88].

In addition, according to our data, a number of miRNAs were significantly decreased only when comparing EVs from the uterine aspirates of OC patients and healthy controls without statistically significant changes in EVs from other body fluids compared to the same control. However, these changes are very pronounced and may also be of significant diagnostic value. Among them there are also known tumor suppressor miRNAs that stimulate apoptosis and the chemotherapy sensitivity of OC cells, such as miR-497-3p and -5p [99,107] and miR-381-3p [108].

## 5. Conclusions

In conclusion, we have identified a panel of 29 miRNAs that exhibit the most pronounced differential co-expression in EVs from all OC-related biofluids compared to normal controls. These results provide the reasonability for the next steps of the study, including the validation of the differential expression of the identified miRNAs in independent patient cohorts by RT-qPCR, as well as the investigation of their functional roles in OC biology. Although we reduced the final panel, it is important to note that some other DE-miRNAs that were excluded from the final list may also be of significant interest in the context of OC diagnosis, especially considering the aforementioned importance of some of them in carcinogenesis. Specifically, those 40 miRNAs that showed similar co-directed changes in EVs from the uterine aspirates of OC patients and other OC-related biofluids should be examined.

## Figures and Tables

**Figure 1 pharmaceutics-16-00902-f001:**
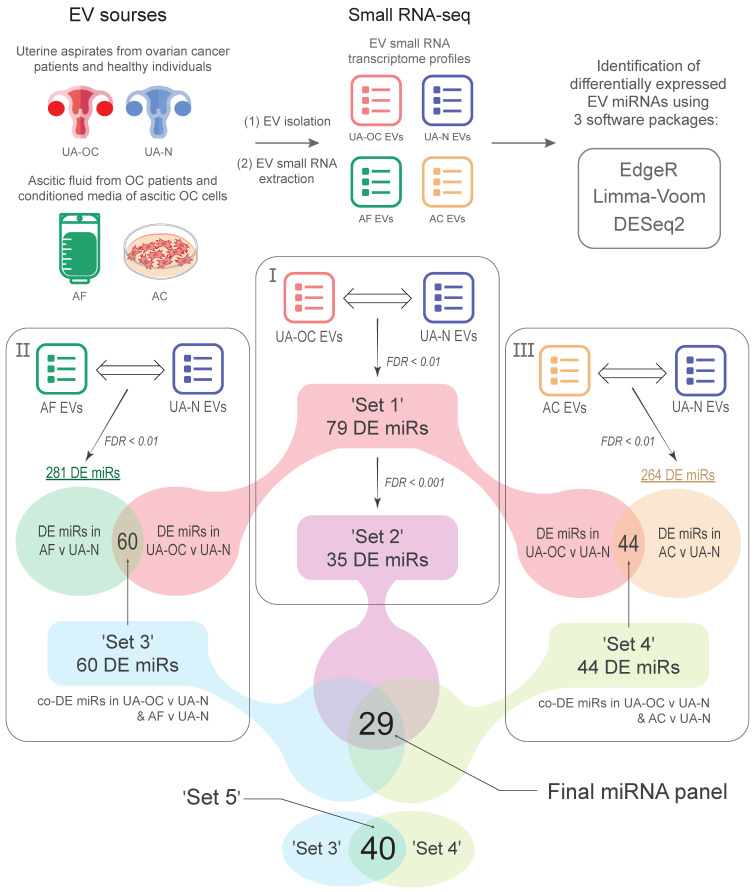
A schematic of the study design. The numbers I-III represent the stages of bioinformatic analysis. EV, extracellular vesicles; UA, uterine aspirate; OC, ovarian cancer; UA-OC, UA from OC patient; UA-N, UA from healthy donor; AF, ascitic fluid; AC, ascitic cells; DE, differential expression.

**Figure 2 pharmaceutics-16-00902-f002:**
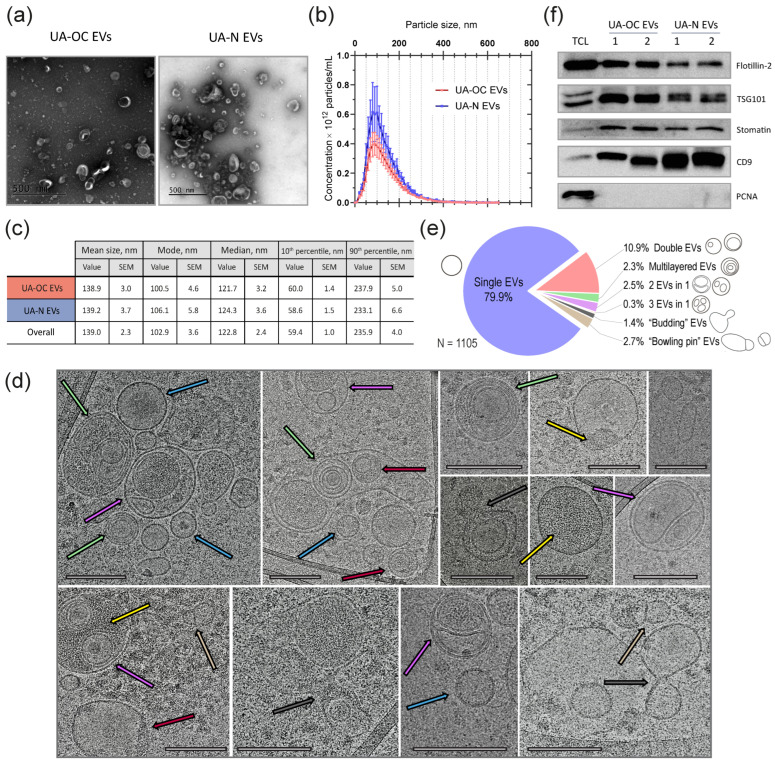
Characterization of EVs isolated from uterine aspirates of OC patients (UA-OC EVs) and healthy donors (UA-N EVs). (**a**) The TEM images show particles with the size and morphology typical of EVs, as well as the presence of atypical vesicles. Scale bars 500 nm. (**b**) NTA data on EV diameter distribution of a population of UA-OC and UA-N samples. Error bars indicate SEM. (**c**) Mean values for EV size characteristics over the entire sample according to NTA data. (**d**) Cryo-EM images illustrating the morphological diversity of EVs. Micrographs of single (blue arrows), double (violet arrows), multilayered (green), “inserted” (red arrows), budding-like (gray) or dumbbell-shaped (light-brown) vesicles, and vesicles with electron-dense cargo (yellow arrows) are shown. Scale bars 200 nm. (**e**) The prevalence of each EV morphological subcategory. N = sample size. (**f**) Western blot analysis of exosomal markers flotillin-2, TSG101, stomatin and CD9 in UA-OC and UA-N EVs. Tumor cell lysate (TCL) obtained from the primary culture of ascites cells was used as a control. The PCNA protein was used to confirm the absence of cellular proteins of non-vesicular origin in EV preparations.

**Figure 3 pharmaceutics-16-00902-f003:**
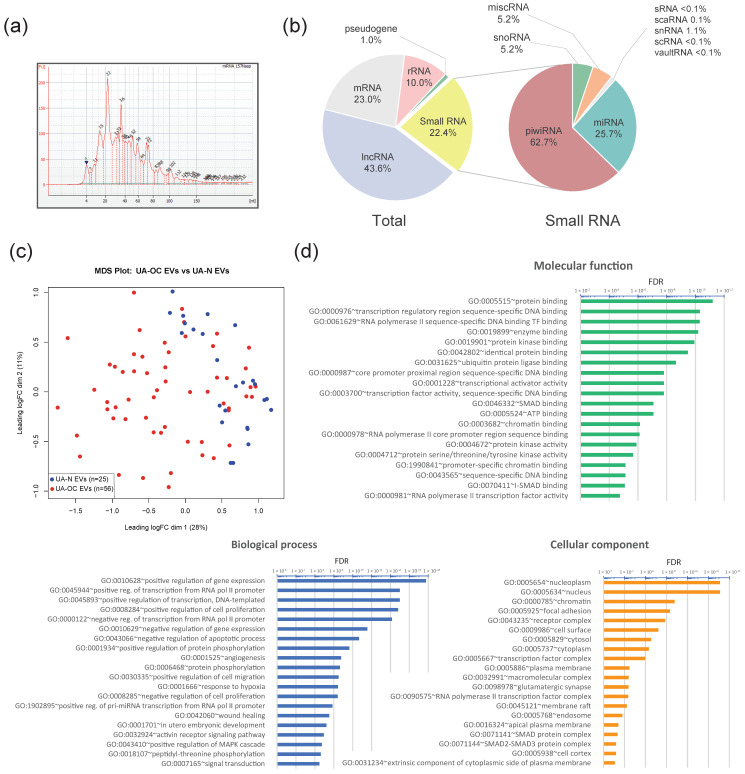
Profiling of small RNAs from UA-OC and UA-N EVs. (**a**) An example of Agilent 2100 Bioanalyzer electropherogram of small RNAs extracted from UA EVs. (**b**) RNA content of the UA EVs according to small RNA-seq data. The mean share percentage of different classes of small RNAs and the total proportion of small RNAs are presented. (**c**) Multidimensional scaling (MDS) plot of miRNA expression profiles among EVs from uterine aspirates of OC patients (red) and non-cancer patients (blue). (**d**) Gene ontology (GO) enrichment analysis of differentially expressed miRNAs from ‘Set 2’ (DE miRNAs meeting the criteria of FDR < 0.001). The top 20 items with the highest FDR are presented.

**Figure 4 pharmaceutics-16-00902-f004:**
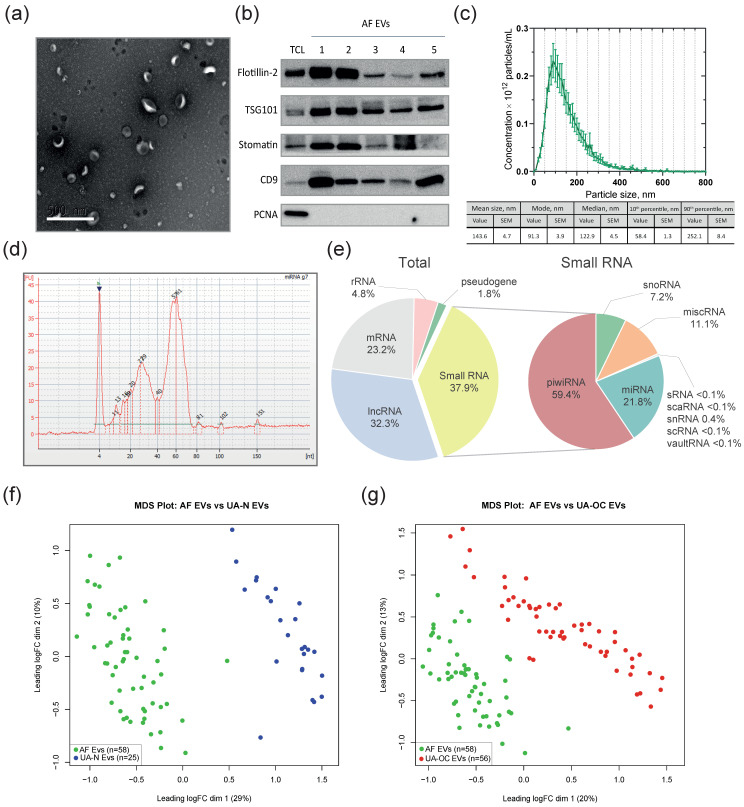
Characterization of EVs isolated from ascitic fluid of OC patients (AF EVs) and their small RNA content. (**a**) A representative TEM image of AF EVs with typical size and cup-shaped morphology. Scale bar 500 nm. (**b**) Western blot analysis of the exosomal markers flotillin-2, TSG101, stomatin, and CD9 in AF EVs. Tumor cell lysate (TCL) obtained from primary culture of ascitic cells was used as a control. The PCNA protein was used to confirm the absence of cellular proteins of non-vesicular origin in EV samples. (**c**) NTA data on EV diameter distribution of a population of AF-derived preparations and mean values for EV size characteristics over the entire sample. Error bars indicate SEM. (**d**) An example of an Agilent 2100 Bioanalyzer electropherogram of small RNAs extracted from AF EVs. (**e**) RNA content of the AF EVs according to small RNA-seq data. The mean share percentage for different RNA types is presented. (**f**) Multidimensional scaling (MDS) plot of miRNA expression profiles among samples of ascitic fluid-derived EVs (AF EVs) (green) and EVs from the uterine aspirates of non-cancer patients (UA-N EVs) (blue). (**g**) MDS plot of miRNA expression profiles among samples of ascitic fluid-derived EVs (AF EVs) (green) and EVs from the uterine aspirates of OC patients (UA-OC EVs) (red).

**Figure 5 pharmaceutics-16-00902-f005:**
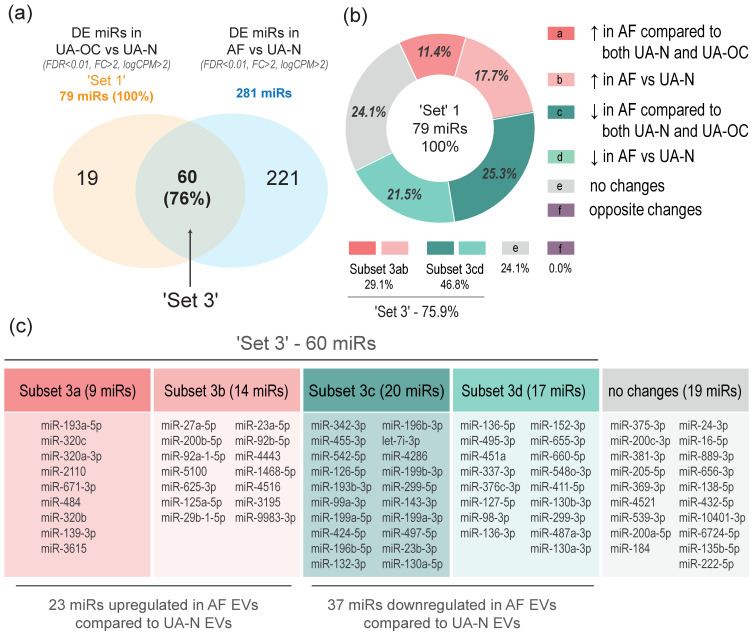
Comparison of miRNAs differentially expressed in EVs from ascitic fluid (AF EVs) and the uterine aspirates of OC patients (UA-OC EVs) versus non-cancer patients (UA-N EVs). (**a**) Venn diagram showing the intersection of two DE miRNA panels, a panel of miRNAs differentially expressed in UA-OC EVs compared to UA-N EVs (‘Set 1’, light yellow) and a panel of miRNAs differentially expressed in AF EVs compared to the same control group (UA-N EVs) (blue). The set of overlapping miRNAs (differentially co-expressed miRNAs) is labeled ‘Set 3’. (**b**) Subsets of the ‘Set 3’ miRNAs, showing differential co-expression when comparing AF EVs with UA-OC and UA-N EVs: subsets “3a” and “3c” correspond to miRNAs differentially co-expressed in AF EVs compared to both UA-N and UA-OC EVs (“3a”—upregulated (in dark red), “3c”—downregulated (in dark green)); subsets “3b” and “3d” correspond to miRNAs differentially co-expressed in AF EVs compared to UA-N EVs only (“3b”—upregulated (in light red), “3d”—downregulated (in light green)); Section “e” (in gray) corresponds to DE miRNAs from ‘Set 1’ that show no differential co-expression in AF EVs, i.e., no significant changes when comparing AF EVs with UA-N EVs; Section “f” shows the absence of DE miRNAs from ‘Set1’ that are significantly changed in the opposite direction in AF EVs. Direction of arrows indicates up- or downregulation of miRNAs. (**c**) List of miRNA molecules corresponding to the indicated subsets of ‘Set 3’ as well as Section “e”.

**Figure 6 pharmaceutics-16-00902-f006:**
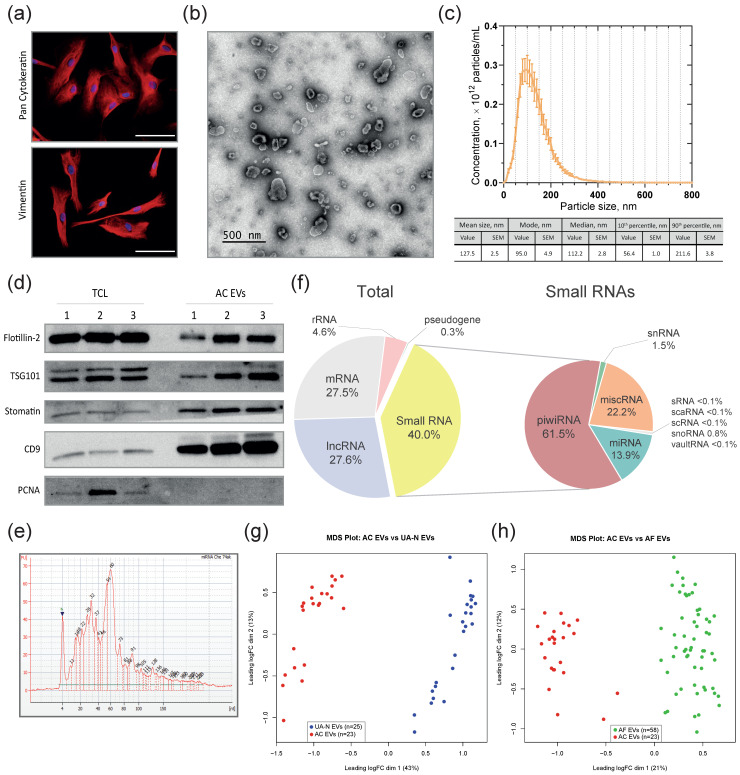
Characterization of EVs isolated from conditioned medium of OC ascites cells in primary culture (AC EVs) and their miRNA content. (**a**) An example of pan cytokeratin and vimentin immunofluorescence staining of cells extracted from malignant ascites and grown in primary culture. Scale bar 100 μm. (**b**) A representative TEM image of AC EVs with the typical size and cup-shaped morphology. Scale bar 500 nm. (**c**) NTA data on EV diameter distribution of a population of AC-derived preparations and the mean values for EV size characteristics over the entire sample of AC EVs according to NTA data. Error bars indicate SEM. (**d**) Western blot analysis of the exosomal markers flotillin-2, TSG101, stomatin and CD9 in AC EVs. Tumor cell lysates (TCL) obtained from parental ascitic cells in primary cultures were used as controls. The PCNA protein was used to confirm the absence of cellular proteins of non-vesicular origin in EV samples. (**e**) An example of an Agilent 2100 Bioanalyzer electropherogram of small RNAs extracted from AC EVs; (**f**) RNA content of AC EVs according to small RNA-seq data. The mean share percentage for different RNA types is presented. (**g**,**h**) MDS plots of miRNA expression profiles in EVs from an AC-conditioned medium (in red) compared to (**g**) UA-N EVs (in blue), and (**h**) AF EVs (in green).

**Figure 7 pharmaceutics-16-00902-f007:**
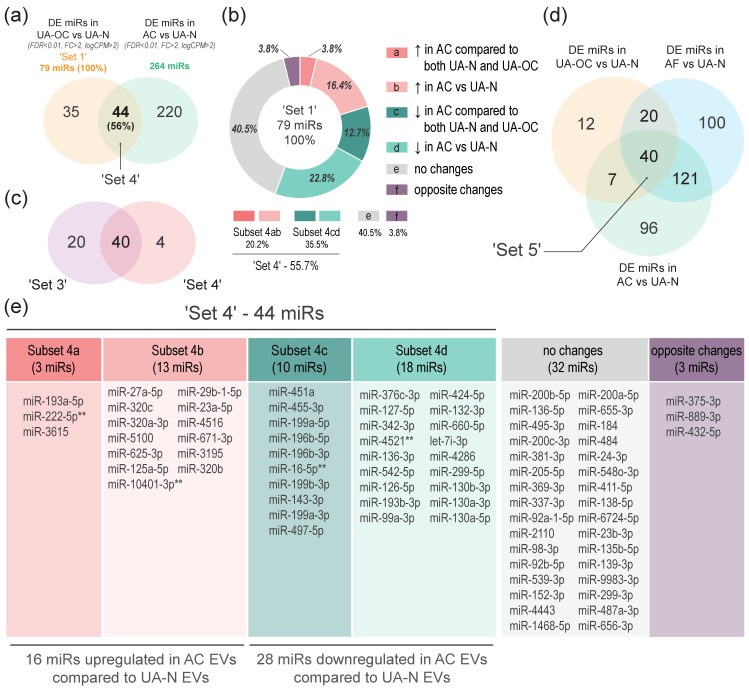
Comparison of miRNAs differentially expressed in EVs from cultured ascites cells, ascitic fluid and the uterine aspirates of OC patients versus non-cancer patients. (**a**) The Venn diagram illustrates the overlap between two DE miRNA panels. The first set, labeled as ‘Set 1’ in light yellow, includes miRNAs that exhibit differential expression in EVs from the uterine aspirates of OC patients (UA-OC EVs) compared to non-cancer patients (UA-N EVs). The second panel, labeled in green, consists of miRNAs that are differentially expressed in EVs from ascites cells grown in primary culture (AC EVs) compared to the same control group (UA-N EVs). The miRNAs that overlap between the two panels exhibit differential co-expression and are labeled as ‘Set 4’. (**b**) Subsets of the ‘Set 4’ miRNAs, showing differential co-expression when comparing AC EVs with UA EVs from OC and non-cancer patients: Subsets “3a” and “3c” correspond to miRNAs differentially co-expressed in AC EVs compared to both UA-N EVs and UA-OC EVs (“3a”—upregulated (in dark red), “3c”—downregulated (in dark green)); subsets “3b” and “3d” correspond to miRNAs differentially co-expressed in AC EVs compared to UA-N EVs only (“3b”—upregulated (in light red), “3d”—downregulated (in light green)). Section “e” (in gray) corresponds to DE miRNAs from ‘Set 1’ that show no differential co-expression in AC EVs, i.e., no significant changes when comparing AC EVs with UA-N EVs. Section “f” displays miRNAs that were significantly altered in AC EVs compared to the control but changed in the opposite direction compared to UA-OC EVs. Direction of arrows indicates up- or downregulation of miRNAs. (**c**) The Venn diagram demonstrating overlapped miRNAs between ‘Set 3’ and ‘Set 4’. (**d**) The Venn diagram illustrating the overlap of three sets of DE miRNAs obtained by comparing UA-OC EVs, AF EVs and AC EVs with UA-N EVs. ‘Set 5’ corresponds to miRNAs significantly altered in the same direction (differentially co-expressed) in EVs from all three biofluids compared to the control. (**e**) The list of miRNA molecules corresponding to the indicated subsets of ‘Set 4’, section “e” and section “f”. Asterisks indicate miRNAs not present in ‘Set 5’.

**Table 1 pharmaceutics-16-00902-t001:** Clinical and morphological characteristics of OC patients (N = 56).

Histology	Serous Carcinoma	Endometrioid Carcinoma	Mucinous Carcinoma	Clear Cell Carcinoma
Cases N	48	5	1	2
Stage				
I	2 (4%)	2 (40%)	-	1 (50%)
II	2 (4%)	-	-	-
III	34 (71%)	3 (60%)	1 (100%)	1 (50%)
IV	10 (21%)	-	-	-
Grade				
G1	8 (17%)	1 (20%)	-	-
G2	3 (6%)	3 (60%)	-	1 (50%)
G3	36 (75%)	1 (20%)	1 (100%)	-
G_unknown_	1 (2%)	-	-	1 (50%)

**Table 2 pharmaceutics-16-00902-t002:** ‘Set 2’ represents miRNAs differentially expressed in UA-OC based on the results obtained using EdgeR ver.3.36.0, Limma-voom ver.3.50.1, and DESeq2 ver.2.11.40.8 software packages, with FDR < 0.001 and FC > 2 criteria. LogFC—base-2 logarithm of the relative change in miRNA levels; logCPM—logarithm of the number of reads per million for a given miRNA; LR—value of statistical significance in the likelihood ratio test; *p*-value—uncorrected *p* value; FDR—adjusted *p*-value for multiple testing; FC (fold change)—recalculated change in miRNA expression. The values obtained from the edgeR package are provided, while the values from the other two packages can be found in the Appendix A. The color red indicates miRNAs that are upregulated in UA-OC compared to UA-N, while the color green indicates miRNAs that are downregulated.

miRNAs Upregulated in UA-OC Compared to UA-N								miRNAs Downregulated in UA-OC Compared to UA-N						
miRNA	logFC	logCPM	LR	*p*-Value	FDR	FC		miRNA	logFC	logCPM	LR	*p*-Value	FDR	FC
miR-27a-5p	4.237	8.517	60.574	7.09 × 10^−15^	9.56 × 10^−12^	18.857		miR-495-3p	−1.904	7.556	40.765	1.72 × 10^−10^	4.64 × 10^−8^	0.267
miR-200b-5p	2.840	8.773	43.317	4.65 × 10^−11^	3.14 × 10^−8^	7.161		miR-381-3p	−1.824	6.177	37.324	1.00 × 10^−9^	1.88 × 10^−7^	0.282
miR-375-3p	2.438	11.469	41.710	1.06 × 10^−10^	3.57 × 10^−8^	5.420		miR-369-3p	−2.102	6.561	31.711	1.79 × 10^−8^	2.20 × 10^−6^	0.233
miR-200c-3p	1.823	12.823	39.722	2.93 × 10^−10^	6.59 × 10^−8^	3.538		miR-451a	−2.672	12.793	30.373	3.57 × 10^−8^	3.33 × 10^−6^	0.157
miR-193a-5p	2.718	7.332	37.110	1.12 × 10^−9^	1.88 × 10^−7^	6.579		miR-337-3p	−1.643	6.122	29.255	6.34 × 10^−8^	5.35 × 10^−6^	0.320
miR-205-5p	3.358	10.035	35.724	2.27 × 10^−9^	3.41 × 10^−7^	10.257		miR-376c-3p	−1.944	3.657	27.124	1.91 × 10^−7^	1.29 × 10^−5^	0.260
miR-92a-1-5p	3.339	3.944	28.781	8.10 × 10^−8^	6.43 × 10^−6^	10.119		miR-127-5p	−1.676	7.121	26.997	2.04 × 10^−7^	1.31 × 10^−5^	0.313
miR-2110	2.480	4.822	28.131	1.13 × 10^−7^	8.50 × 10^−6^	5.577		miR-4521	−1.672	4.424	22.549	2.05 × 10^−6^	8.92 × 10^−5^	0.314
miR-5100	5.146	5.276	27.980	1.23 × 10^−7^	8.71 × 10^−6^	35.400		miR-98-3p	−1.555	5.206	22.025	2.69 × 10^−6^	1.04 × 10^−4^	0.340
miR-625-3p	1.567	6.313	26.049	3.33 × 10^−7^	1.95 × 10^−5^	2.962		miR-539-3p	−2.135	5.598	20.607	5.64 × 10^−6^	1.77 × 10^−4^	0.228
miR-125a-5p	1.830	13.350	23.467	1.27 × 10^−6^	6.13 × 10^−5^	3.556		miR-136-3p	−1.753	7.573	20.357	6.42 × 10^−6^	1.97 × 10^−4^	0.297
miR-671-3p	1.764	5.307	23.134	1.51 × 10^−6^	7.04 × 10^−5^	3.396		miR-152-3p	−1.188	10.297	20.020	7.66 × 10^−6^	2.30 × 10^−4^	0.439
miR-29b-1-5p	2.515	3.417	22.098	2.59 × 10^−6^	1.03 × 10^−4^	5.718		miR-542-5p	−1.700	4.504	19.259	1.14 × 10^−5^	3.06 × 10^−4^	0.308
miR-23a-5p	3.774	3.797	21.943	2.81 × 10^−6^	1.04 × 10^−4^	13.682		miR-655-3p	−1.516	5.976	19.235	1.16 × 10^−5^	3.06 × 10^−4^	0.350
miR-484	1.278	9.722	16.749	4.27 × 10^−5^	8.73 × 10^−4^	2.425		miR-193b-3p	−1.176	7.429	17.635	2.68 × 10^−5^	6.12 × 10^−4^	0.443
								miR-99a-3p	−1.082	6.079	17.541	2.81 × 10^−5^	6.33 × 10^−4^	0.473
								miR-199a-5p	−1.093	14.674	17.337	3.13 × 10^−5^	6.82 × 10^−4^	0.469
								miR-130a-5p	−1.780	4.420	25.248	5.04 × 10^−7^	2.72 × 10^−5^	0.290
								miR-136-5p	−2.468	5.759	42.428	7.33 × 10^−11^	3.30 × 10^−8^	0.181
								miR-487a-3p	−1.535	3.802	15.046	1.05 × 10^−5^	9.63 × 10^−4^	0.345

**Table 3 pharmaceutics-16-00902-t003:** The 35 miRNAs with the highest differential expression (FDR < 0.001) in UA-OC EVs (‘Set 2’) and their overlap with DE miRNAs from ‘Set 3’ and ‘Set 4’. The FDR values obtained from the edgeR package are provided; n/s—non-significant. MiRNAs exhibiting co-directional changes in all three biofluids investigated are indicated in dark orange (if upregulated) and dark blue (if downregulated). Light shading indicates miRNAs with co-directional changes observed in two biofluids.

DE miRs in UA-OC vs UA-N	DE miRs in AF vs UA-N		DE miRs in AC vs UA-N			DE miRs in UA-OC vs UA-N	DE miRs in AF vs UA-N		DE miRs in AC vs UA-N	
*(FDR < 0.001)*	*(FDR < 0.01)*		*(FDR < 0.01)*			*(FDR < 0.001)*	*(FDR < 0.01)*		*(FDR < 0.01)*	
‘Set 2’	‘Set 3’		‘Set 4’			‘Set 2’	‘Set 3’		‘Set 4’	
UPREGULATED	UP or DOWN	FDR	UP or DOWN	FDR		DOWNREGULATED	UP or DOWN	FDR	UP or DOWN	FDR
miR-27a-5p	↑	4.27 × 10^−8^	↑	2.84 × 10^−15^		miR-451a	↓	7.16 × 10^−9^	↓	9.19 × 10^−27^
miR-193a-5p	↑	1.85 × 10^−4^	↑	2.53 × 10^−28^		miR-376c-3p	↓	2.29 × 10^−7^	↓	5.65 × 10^−3^
miR-5100	↑	2.99 × 10^−17^	↑	1.57 × 10^−16^		miR-127-5p	↓	4.96 × 10^−10^	↓	6.61 × 10^−3^
miR-625-3p	↑	2.55 × 10^−8^	↑	7.07 × 10^−7^		miR-136-3p	↓	3.43 × 10^−3^	↓	3.03 × 10^−5^
miR-125a-5p	↑	9.78 × 10^−5^	↑	2.58 × 10^−3^		miR-542-5p	↓	1.44 × 10^−20^	↓	8.87 × 10^−16^
miR-671-3p	↑	2.23 × 10^−18^	↑	2.90 × 10^−12^		miR-193b-3p	↓	1.37 × 10^−9^	↓	1.42 × 10^−3^
miR-29b-1-5p	↑	5.92 × 10^−6^	↑	1.70 × 10^−7^		miR-99a-3p	↓	1.88 × 10^−25^	↓	4.61 × 10^−35^
miR-23a-5p	↑	8.50 × 10^−15^	↑	2.52 × 10^−6^		miR-199a-5p	↓	6.82 × 10^−26^	↓	4.95 × 10^−77^
miR-200b-5p	↑	1.15 × 10^−7^		n/s		miR-130a-5p	↓	6.60 × 10^−15^	↓	5.18 × 10^−11^
miR-92a-1-5p	↑	4.13 × 10^−8^		n/s		miR-4521		n/s	↓	6.91 × 10^−10^
miR-2110	↑	5.80 × 10^−24^		n/s		miR-136-5p	↓	5.59 × 10^−5^		n/s
miR-484	↑	2.33 × 10^−16^		n/s		miR-495-3p	↓	1.01 × 10^−17^		n/s
miR-375-3p		n/s		n/s		miR-337-3p	↓	5.10 × 10^−13^		n/s
miR-200c-3p		n/s		n/s		miR-98-3p	↓	5.26 × 10^−9^		n/s
miR-205-5p		n/s		n/s		miR-152-3p	↓	8.95 × 10^−3^		n/s
						miR-655-3p	↓	4.36 × 10^−4^		n/s
						miR-487a-3p	↓	2.07 × 10^−4^		n/s
						miR-381-3p		n/s		n/s
						miR-369-3p		n/s		n/s
						miR-6393p		n/s		n/s

## Data Availability

The data that supports the findings of this study are available in the Appendix A of this article. The small RNA-seq data that support the findings of this study are openly available in the NCBI Sequence Read Archive (SRA) under project number PRJNA1070217 (https://www.ncbi.nlm.nih.gov/sra/PRJNA1070217).

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
