# Peer review of "Integrated miRNA Profiling of Extracellular Vesicles from Uterine Aspirates, Malignant Ascites and Primary-Cultured Ascites Cells for Ovarian Cancer Screening"

_pharmaceutics, 2024, doi:10.3390/pharmaceutics16070902_

Round 1

Reviewer 1 Report

Comments and Suggestions for Authors

In the reviewed manuscript, Skryabin et al present detailed characterizations of the micro RNA profiles isolated from extracellular vesicles of uterine aspirates, malignant ascites and cultured ascites cells of ovarian cancer patients and compare it with the profile of the one prepared from a non- malignant control group. Most of the necessary experiments to achieve this aim  have been carried out and described in this work. Nevertheless, it is difficult to follow the experimental design and to distinguish the exact differences between the groups. Therefore I would recommend to include an a graphical representation of the comparisons made. For a work in which the main focus is to characterize vesicles from differently processed samples from ovarian cancer patients, it would make sense to include a comparative representation of size, concentration, shape as shown in Figure1 for all the samples. Additionally a reconstruction of the introduction would make sense. The last paragraph would fit perfectly as first paragraph. The discussion is very long and contains many repetitions of the remaining results, which is not absolutely necessary. In my opinion, it would make sense to streamline this section. In summary, after the suggested improvements the manuscript is worth to be published.  

Comments on the Quality of English Language

The Quality of the English language is appropriate

Author Response

We thank the reviewer for the careful analysis of the study, comments and suggestions to improve the quality of the manuscript. According to them, we have made changes in all the points mentioned in the review. 

1.Therefore I would recommend to include an a graphical representation of the comparisons made.

Thank you for the recommendation. We wanted to make some kind of schematic ourselves, but it turned out to be very complicated every time. This time we think we were able to come up with a pretty good graphical representation of the experimental design and therefore made it the first figure in the manuscript. We hope that this will really help the perception of the study as a whole (see Figure 1).

2.For a work in which the main focus is to characterize vesicles from differently processed samples from ovarian cancer patients, it would make sense to include a comparative representation of size, concentration, shape as shown in Figure1 for all the samples.

Thank you for your comment. We have added a table comparing the mean values of vesicle size and concentration in all biofluids: uterine aspirates, ascites, and conditioned medium from primary tumor cell cultures. Since the comparison showed no differences in the characteristics studied between these vesicle groups, we have placed the table in the Supplementary section so as not to overload the main text of the manuscript. (see Supplementary Table S3).

3.Additionally a reconstruction of the introduction would make sense. The last paragraph would fit perfectly as first paragraph.

Thank you for your comment. We took your advice and swapped out those text fragments. It really looks better that way (see lines 32-36).

 4.The discussion is very long and contains many repetitions of the remaining results, which is not absolutely necessary.

Thank you for the recommendation.  In response to your comments, we have shortened some parts of the discussion. However, since other reviewers liked the discussion in present form and found it important for understanding the results in general, we did not change this part much. In addition, also based on comments from other reviewers, we even had to slightly expand this section by adding a discussion of the role of some more miRNAs in ovarian cancer carcinogenesis.

Reviewer 2 Report

Comments and Suggestions for Authors

Integrated miRNA profiling of extracellular vesicles from uterine aspirates, malignant ascites and primary cultured ascites cells for ovarian cancer screening

Abstract: 

Overall, this is a well-designed and thoughtfully executed study that leverages an integrative approach to identify a refined panel of extracellular vesicle (EV)-associated miRNAs as potential biomarkers for ovarian cancer screening. The authors are commended for their careful collection and processing of clinically relevant biofluid samples, as well as their systematic bioinformatics analysis to uncover differentially expressed miRNAs across multiple ovarian cancer-related sample types.

Clinical samples 

Overall, this section outlines the careful collection and handling of clinical samples from well-characterized patient cohorts, adhering to appropriate ethical guidelines, which lends credibility to the subsequent analysis and findings. However, author should include the ethical approval number for more accuracy. 

Specific comments:

Sample collection and processing:

The authors provide a detailed account of the clinical sample collection and handling procedures, which adheres to appropriate ethical guidelines. It would be helpful to include the specific ethical approval number for transparency.

The sample processing workflow, including EV isolation by differential ultracentrifugation, aligns with commonly used methods. However, the authors could strengthen their justification for choosing this approach over more advanced techniques, such as immunomagnetic separation, which offer potential advantages in terms of selectivity, efficiency, and preservation of EV characteristics.

It would be useful for the authors to provide a citation or reference to the original source that informed their sample processing methodology, including the cell culture procedures.

Bioinformatics analysis:

The authors' use of three independent software packages to identify differentially expressed miRNAs (DE-miRNAs) in uterine aspirate EVs is a robust approach that enhances the confidence in the findings.

The comparative analysis of DE-miRNAs across the different biofluid sources (uterine aspirates, ascites fluid, and ascites cells) is a key strength of the study, as it allows for the identification of a refined panel of miRNAs with consistent differential expression.

The authors are encouraged to discuss the potential biological and functional significance of the identified 29-miRNA panel, if known, and how these miRNAs may relate to ovarian cancer pathogenesis or serve as useful screening biomarkers.

Limitations and future directions:

The authors provide a comprehensive overview of the key limitations inherent to miRNA profiling studies, including the lack of standardization, small sample sizes, tissue/cell specificity, and challenges in validating the functional relevance of identified miRNAs.

Addressing these limitations through improved study designs, larger sample sizes, standardized protocols, and integration with other omics data could enhance the insights and translational potential of this line of research.

The authors may consider discussing potential next steps, such as validating the identified miRNA panel in independent patient cohorts, characterizing the surface markers of EVs from different sources, and investigating the functional roles of the candidate miRNAs in ovarian cancer biology.

In conclusion, this is a well-executed study that contributes to the growing body of evidence on the potential of EV-associated miRNAs as biomarkers for ovarian cancer. The authors' integrative approach and thoughtful discussion of the limitations and future directions provide a solid foundation for further investigation in this important research area. With the suggested improvements, this manuscript has the potential to be a valuable reference for similar studies and garner significant citation in the field.

Author Response

We thank the reviewer for the high evaluation of our study, careful analysis, comments and suggestions to improve the quality of the manuscript. According to them, we have made changes in all the points mentioned in the review.

 1.However, author should include the ethical approval number for more accuracy

Thank you for your comment. The Ethics Committee of N.N. The Ethics Committee of Blokhin National Medical Research Center of Oncology issues protocols of not quite standard format, including two documents - permission to collect clinical material and confirmation that the data presented in the manuscript were obtained in accordance with the permission. The numbers are different, but we have provided all the information (see lines 155-156), and more detailed information has been submitted to the Editorial Board

  1. 2. However, the authors could strengthen their justification for choosing this approach over more advanced techniques, such as immunomagnetic separation, which offer potential advantages in terms of selectivity, efficiency, and preservation of EV characteristics.

Thank you for your comment. Vesicle isolation methods based on ultracentrifugation are well established and recommended by ISEV, although there are many other methods available. As for immunosorption-based methods, including immunomagnetic separation, their use a priori leads to a narrowing of the pool of vesicles to be isolated, since single molecules considered as markers of EVs are used for isolation. According to current knowledge, each of these markers may be poorly represented or even absent on individual vesicle populations, while other markers may be highly represented in the same vesicles. Other populations of EVs in the same sample may be characterized by a different ratio of markers. For this reason, ISEV requires the mandatory analysis of multiple markers, as the presence or absence of one marker alone cannot confirm or deny the presence of vesicles in the preparations. Therefore, precipitation of EVs with single proteins could not isolate the entire pool of EVs. That's why we prefer differential centrifugation.

  1. It would be useful for the authors to provide a citation or reference to the original source that informed their sample processing methodology, including the cell culture procedures.

Thank you for your comment. We have added two links to the original articles from which the methods for processing UA samples, as well as techniques for obtaining primary tumor cell cultures and culturing them, were taken (see lines 165-166 and lines 171-172 respectively).

4 The authors are encouraged to discuss the potential biological and functional significance of the identified 29-miRNA panel, if known, and how these miRNAs may relate to ovarian cancer pathogenesis or serve as useful screening biomarkers.

Thank you for your comment. We have added more miRNAs to the discussion, especially miR-136, and also some additional information about miR-495-3p and miR-152. Unfortunately, it is not possible to discuss all 29 miRNAs in this context due to the already large amount of discussion (perhaps a separate mini-review focusing on the biological and functional significance of the identified miRNA panel and their potential as OC markers would be useful in the future) (see lines 939-952).

5.The authors may consider discussing potential next steps, such as validating the identified miRNA panel in independent patient cohorts, characterizing the surface markers of EVs from different sources, and investigating the functional roles of the candidate miRNAs in ovarian cancer biology.

Thank you for your comment. First of all, we would like to point out that this is exactly what we are currently doing (!), including collecting samples from an independent cohort of patients, validating the identified miRNAs by RT-qPCR, and investigating the functional roles of the candidate miRNAs in ovarian cancer biology! Following your comment, we have added this information, but not in the Discussion section, but in the Conclusion (see lines 981-984).

Reviewer 3 Report

Comments and Suggestions for Authors

Skryabin et al have analysed the RNA content of EVs derived from uterine aspirates collected from patients with ovarian cancer and unaffected controls. They have focussed on changes in miRNA cargo, generating a list of differentially packaged miRNAs, which could be of relevance in ovarian cancer diagnosis. Additionally, the list was refined by comparison with ascites fluid EVs and EVs isolated from media collected from cultured ascites cells. The authors have characterised the EVs to ISEV standards and have performed detailed bioinformatic analysis to reach their final list of miRNAs. Potential questions that could be raised regarding the comparisons of AF and AC EVs to the UA normal controls have been addressed by the authors, which is good to see. They have also commented on the fact that samples were treated as similarly as possible to avoid variation in EV cargo resulting from sample processing.

I think the data is presented well and the conclusions are balanced. I only have some minor queries/comments.

Minor points:

1. Lines 421-427 describe how Set 1 miRNAs altered between different grades of disease. Interestingly, the two miRNAs described are not seen in the Set 5 list. Maybe the authors could comment on this?

2. Lines 307-309 need removed (guidance for writing results)

3. Line 413 has a full hyperlink for the reference rather than reference number.

4. Line 667 - Corresponding changes in miRNA levels are observed in the row “UA-N EVs – UA-OC EVs– AF EVs – AC EVs”. I am not sure where this row is shown? Similar rows mentioned in lines 831 and 834. The authors should specify where in the manuscript to find the rows they are referring to.

Author Response

We thank the reviewer for the analysis of our study, comments and suggestions to improve the quality of the manuscript. According to them, we have made changes in all the points mentioned in the review.

  1. Lines 421-427 describe how Set 1 miRNAs altered between different grades of disease. Interestingly, the two miRNAs described are not seen in the Set 5 list. Maybe the authors could comment on this?

Thank you for your comment. We believe that your question arose due to our unclear wording. Since miR-3180-3p and miR-383-5p are not actually part of Set1, i.e. they do not show differential expression when comparing uterine aspirates from OC patients and healthy donors (i.e. UA-OC vs. UA-N). Differential expression of these miRNAs is only detected when comparing OC samples among themselves, i.e. when comparing miRNA profiles in groups of high-grade and low-grade OC. Accordingly, these miRNAs had "no chance" to be in set 5, which is the intersection of set 1 with other sets. To avoid questions, we made a more precise statement: "Within the group of UA-OC samples, miRNA profiles were then compared according to clinical and morphological characteristics of the tumors, such as histological type, extent and stage of disease" (see lines 463-465). It should be noted that low-grade and high-grade OC are such different groups that they are now even considered to be different subtypes of the disease, with probably different etiologic factors and possibly different molecular signatures. Therefore, the results obtained rather indicate the specificity of miR-3180-3p and miR-383-5p and the prospect of their use as markers for the differential diagnosis of OC. We are planning to test this possibility using OT-PCR in an independent cohort. In this case (to return to your question), we are also interested in comparing the G1 and G2/G3 groups not only with each other, but also with the norm. We may be able to see some differences.

  1. Lines 307-309 need removed (guidance for writing results)

Thank you for your comment. This is a technical error. We have removed this sentence.

  1. Line 413 has a full hyperlink for the reference rather than reference number.

Thank you for the comment, the link has been corrected.

  1. Line 667 - Corresponding changes in miRNA levels are observed in the row “UA-N EVs – UA-OC EVs– AF EVs – AC EVs”. I am not sure where this row is shown? Similar rows mentioned in lines 831 and 834. The authors should specify where in the manuscript to find the rows they are referring

Thank you for your comment!  Again, we believe that your question arose from our unclear wording, specifically the inaccurate use of the word "rows". We have replaced it with "series" and added clarifying sentences to make it clear what we are talking about.

Below are three pieces of text with the corrections made.

“Since most of the molecules from subsets 4a and 3b are also present in subsets 3a and 3b, this means that their levels in body fluids change in the same direction (increase or decrease) as they become more relevant to OC. In other words, the levels of such miRNAs consistently increase or decrease in the series “UA-N EVs – UA-OC EVs– AF EVs – AC EVs”.” (see lines 690-694)

“The next set consists of 60 miRNAs co-differentially expressed in AF EVs and UA-OC EVs compared to UA-N EVs ('Set 3'). Among them, 29 miRNAs show a co-directed changes in the series “UA-N – UA-OC – AF” (“Subsets 3a/3c”). The last one ('Set 4') is represented by 44 miRNAs that are co-expressed in AC EVs and UA-OC EVs compared to control UA-N EVs. Among them, 13 miRNAs show a co-directed changes in the series “UA-N – UA-OC – AC” (“Subsets 4a/4c”). Comparing these molecules, it is interesting to highlight the common miRNAs present in both “Subsets 3a/3c” and “Subsets 4a/4c”, i.e. those miRNAs whose differential co-expression becomes more pronounced as the relevance of the biological fluid to ovarian cancer increases (i.e. in the row "UA-N EVs - UA-OC EVs - AF EVs - AC EVs"). These include ten miRNAs: miR-193a-5p, miR-3615, miR-455-3p, miR-199a-5p, miR-199a-3p, miR-199b-3p, miR-196b-5p, miR-196b-3p, miR-143-3p, miR-497-5p. Of particular interest is the set of miRNAs obtained by comparing DE miRNAs from 'Set 3' and 'Set 4” with “Set 2 “(Table 3).” (see lines 884-897)